# Sequencing of the invasive *E. coli* strain BEN2908 isolated from poultry: A comparative investigation of genomic regions shared with intestinal and extraintestinal model *E. coli* strains

Tobias Weber Martins[1], Angélina Trotereau[2], Simone Lahnig-Jacques[1], Maxime Branger[2], Sébastien Houle[3], Charles M. Dozois[3], Daniel Brisotto Pavanelo[1], Fabiana Horn[1]*, Catherine Schouler[2]*

1 Departamento de Biofísica, Universidade Federal do Rio Grande do Sul, Porto Alegre, Rio Grande do Sul, Brazil, 2 INRAE, Université de Tours, ISP, F-37380, Nouzilly, France, 3 INRS—Centre Armand-Frappier Santé Biotechnologie, Laval, Quebec, Canada

* catherine.schouler@inrae.fr (CS); fabiana.horn@ufrgs.br (FH)

## Abstract

Extraintestinal pathogenic *Escherichia coli* (ExPEC) cause disease outside the gut and include avian pathogenic *E. coli* (APEC), a leading cause of bacterial infections in poultry. Among their highly diverse types, strain BEN2908 stands out for its significant invasive ability across various human and avian cell types. Aiming to investigate further aspects of this strain and its plasmid, we sequenced and assembled the complete genome of BEN2908 and compared it to 22 *E. coli* strains, including other invasive strains such as adherent and invasive *E. coli* (AIEC) LF82 and NRG857c, by constructing a phylogenetic tree and using web-based characterization software. With these results, we selected eight strains closely related to BEN2908 to perform a ring comparison, including two APEC (APEC O1 and IMT5155), two neonatal meningitis *E. coli* (NMEC; RS218 and IHE3034), two uropathogenic *E. coli* (UPEC; 78-Pyelo and CFT073), one commensal *E. coli* (MG1655) and one adherent-invasive *E. coli* (AIEC; LF82). This revealed 20 genomic regions (GRs) of interest which were then analysed by CD-Search, BLASTp and KEGG Pathway databases. Many of the genes in these GRs had no previous description but showed similarity to known genes involved in sugar uptake, nitrogen metabolism, and dicarboxylate transport and processing, among other functions. These results were tabulated and used to infer possible pathways that could be involved in ExPEC pathogenesis, highlighting candidate genes that have been overlooked in ExPEC research.

## Introduction

Avian pathogenic *Escherichia coli* (APEC) causes avian colibacillosis, a prevalent bacterial infection that can affect birds of all ages and at all stages of poultry

**Data availability statement:** The sequences of the BEN2908 chromosome and plasmid are available in GenBank under the following accession numbers: LR740776.1; LR740777.2. Raw reads were submitted to Sequencing Read Archive (SRA) under the following BioProject: PRJNA1359407. The Python scripts used, BRIG alignment files, FASTA sequences of each GR, their annotations generated by RAST, as well as EzAAI, RAxML, Orthofinder, Roary, PHASTEST outputs and the sequencing files mentioned in methods section, are available on GitHub at the following repository: https://github.com/Martins-TW/BEN2908_Genome_Analysis.git. BEN2908 strain has been deposited at the International Center for Microbial Resources—Bacterial Pathogens (CIRM-BP) under name CIRMBP-1386.

**Funding:** This study has received funding from DGAL within the EcoAntibio2 call, COLIPHAVI project, France (C.S.), and from CNPq (Projeto Universal 423.902/2016-4) and FAPERGS (PPSUS 21/2551-0000079-1), Brazil (F.H.). T.W.M. was the recipient of a CAPES Master studentship (DS 88887921543/2023-00). A.T. was supported by a training grant from the Fédération de recherche en infectiologie (FéRI). C.M.D. received funding from NSERC Discovery Grant 2019-06642. The funders had no role in the study design, data collection and analysis, decision to publish, or preparation of the manuscript.

**Competing interests:** The authors have declared that no competing interests exist.

production. It manifests in a range of clinical forms, including omphalitis in embryos, salpingitis in laying hens, cellulitis, airsacculitis, perihepatitis, peritonitis, and septicaemia [1]. Avian colibacillosis results in high morbidity and mortality leading to economic losses in the industry throughout the world (e.g., in Netherlands, losses due to salpingitis were estimated at € 0.4 million, € 3.3 million and € 3.7 million for the layer-sector, the meat-sector and poultry farming, respectively [2]). APEC strains are typical commensal inhabitants of the chicken intestine, exhibiting a high genetic diversity, reflected by the various serogroups and sequence types (ST) isolated from clinical cases. To cause an extraintestinal infection, though, an APEC would require siderophores, which permit them to survive in the host fluids (poor in free iron), and protectins (such as K1 capsule), with which they can evade host defenses. Yet, even if some virulence genes/strategies have been identified, the pathophysiology of avian colibacillosis remains incompletely understood, and additional determinants likely remain to be discovered. One such example is the dicarboxylate uptake regulator, DctR, which appears to contribute to biofilm formation, serum resistance, adherence and colonization in ducks [3]. In that perspective, many APEC genomes have been sequenced; to date (December, 2024), of the 349,451 *E. coli* genomes available on Enterobase, 21,666 were isolated from poultry, of which 468 reportedly belong to ST95. ST95 strains exhibit a broad host range, and phylogenetic studies show that strains isolated from different species share numerous genes and SNPs, indicating a substantial degree of genomic overlap [4]. Among *E. coli* strains of this ST, BEN2908 (O2:H5:K1) is one of the APEC strains that has been most studied. Moreover, the ST95 complex (STc95, a ST complex defined as having at least three STs that differ among each other by no more than two of the seven loci) is also closely related to different types of ExPEC infection, being one of the five most common STc in ExPEC, at least for the past few decades [5–8].

Strain BEN2908 is a nalidixic acid resistant derivative of strain MT78, isolated in 1977 from the trachea of a one-day old chick in France [9]. BEN2908 is an efficient colonizer of the chicken intestine [10,11], and its inoculation either by the air sacs or intratracheally results in severe systemic infection [12,13]. In mice, transurethral inoculation of BEN2908 results in urinary infections comparable to those caused by human UPEC strains [14]. Remarkably, BEN2908 is able to metabolize short chain fructooligosaccharides, conferring a competitive advantage to colonize the intestine of chickens [10,11,15]. Carbohydrate metabolism also seems to play a role in colonization of the lungs and/or air sacs by BEN2908 [16]. Genes linked to sugar metabolism also contribute to bacterial fitness under stressful conditions such as oxygen restriction, the late stationary phase of growth, or growth in serum or in the intestinal tract [17,18]. Type 1 fimbriae expression, in particular, seems to be dependent on regular cytosolic levels of carbohydrates [18,19]. Further, this strain can also invade and survive within avian and human cells including avian fibroblasts [20] and hepatocytes (LHM), human pneumocytes (A549) [21], human brain microvascular endothelial cells (HBMEC) [22], and human intestinal cells (Intestine-407), at levels comparable to AIEC strains [23]. Moreover, BEN2908 has also been used as a model to study important ExPEC-specific genes such as *ibeA,* which contributes to

adherence to host cells, resistance to oxidative stress and virulence in both avian and mammalian hosts [22,24,25]. Due to its ability to induce colibacillosis in poultry, cause urinary tract infections in murine models, and invade various types of human cells, BEN2908 represents a highly versatile and cross-host pathogenic *E. coli*.

Another subset of pathogenic *E. coli* includes those belonging to the AIEC pathotype, which are commonly associated with high adhesion and invasion rates causing inflammatory bowel diseases (IBD) in humans, such as ulcerative colitis or Crohn's disease. Strain LF82 is an extensively studied AIEC reference strain and one of the first identified as capable of inducing IBD, followed by AIEC NRG857c, isolated from the ileum of a patient with Crohn's disease [26,27]. In addition to its capacity to invade different intestinal cell lines, these AIEC strains display notable genomic similarity to some ExPEC strains, such as strain APEC O1 and UPEC strain UTI89 [28]. AIEC strains also have some well-known virulence factors that are essential for adherence, invasion and survival inside host cells, such as the serine autotransporter protease Vat, which induces vacuolization and cytoskeleton rearrangements in avian, murine and human cells [29–31]. Moreover, transcriptomics and *Tn*-seq analyses identified LF82-specific genes that might be implicated in the growth or survival of intracellular bacterial communities (IBCs). This investigation uncovered three noteworthy gene clusters: the High Pathogenicity Island (HPI), a putative type 6 secretion system (T6SS), and a region associated with carbohydrate metabolism [31]. Curiously, the functionality of those three clusters has already been correlated to ExPEC, especially APEC, pathogenesis by many different authors [32–36].

In this study, we perform a comparative genomic analysis of BEN2908 to ten ExPEC strains, ten intestinal pathogenic *E. coli* (InPEC) strains, and two commensal strains, to identify genomic features common to these different strains, that are potentially underlying their ability to cause pathogenesis. To achieve that, we: (i) demonstrated the close evolutionary relationship between BEN2908, AIEC LF82 and NRG857c and other ExPEC model strains through a phylogenetic analysis, complemented by different characterization programs, (ii) identified and analysed the content of 20 genomic regions (GRs) in common to these strains that were absent in the commensal *E. coli* K-12, and (iii) inferred the possible origin of some of these features by comparison to analogous genomic modules from other bacteria. Through this approach, our aim was to identify novel genes or gene modules that may have a functional role in *E. coli* pathogenesis due to their homology to other known genes. Moreover, the complete sequence of the chromosome and plasmid, identified, described, and analysed in this study will be useful to study more comprehensively the pathogenic mechanism of strain BEN2908 and other invasive *E. coli*.

## Methods

### BEN2908 DNA extraction, sequencing, assembly and annotation

Nanopore sequencing and assembly were performed by the Genome and Transcriptome Facility at Bordeaux, France. For the extraction, 5 μg of genomic DNA were sheared to 20 Kb using Megaruptor 2 (Diagenode). Sheared DNA was End-Repaired using Oxford Nanopore recommendations for 1D Ligation sequencing (LSK-SQK 108), with minor modifications, as follows. 48 μL of sheared DNA were incubated with 7 μL of Ultra II End-prep reaction buffer and 3.5 μL of Ultra II End-prep enzyme mix (New England Biolabs) at 20 °C for 15 minutes and at 65 °C also for 15 minutes. The sample was then cleaned up using 1.0X of AMPure XP beads and barcoded using NBD-103 kit (Oxford Nanopore Technologies). After that, 22.5 μL of clean repaired DNA and 25 μL of Blunt/TA Ligase Master Mix (New England Biolabs) were added to 2.5 μL of barcode and incubated at room temperature for 15 minutes. Barcoded samples were cleaned up again and quantified using Qubit fluorometer (Invitrogen) for equimolar pooling. Then, 2.2 μg of pooled DNA were ligated to AMX adapter and purified according to Oxford Nanopore recommendations. Afterwards, 13 μL of the library were loaded into a MinION Flow cell (FLO-MIN106 R9.4) and sequenced during 48 hours on a GridION x5. The obtained raw data (.fast5 files) were base called in high accuracy mode and filtered using Guppy (v. 4.0.11) by applying a minimum quality score of Q7. This resulted in 752,742 reads above the Q7 cutoff. Read quality and length distributions were then assessed using NanoPlot [37] and compared with the dataset prior to filtering.

Illumina paired-end sequencing was performed in 2015 at the Genome Québec facility, at McGill University (Montreal, Quebec, Canada) in an Illumina MiSeq machine; sequencing generated 3,274,430 reads with 250 bp of length and 50% GC content. Illumina reads were trimmed for adapters and low-quality bases using Trimmomatic (v. 0.32) [38], with the following parameters: ILLUMINACLIP:TruSeq3-PE.fa:2:30:10; LEADING:30; TRAILING:30; HEADCROP:20; MINLEN:150. This resulted in 2,476,272 paired reads with minimum length of 150 bp, which were further filtered with a Q20 cutoff using fastq_quality_filter (v. 1.0.0), available on Galaxy platform (v. 25.0) [39]. After filtering, a total of 1,713,025 reads were retained, yielding a coverage depth of 282x and breadth of 99.86%, as assessed by BWA (v. 0.7.19) [40]. The files and logs of the programs mentioned above, including the FastQC score (v. 0.12.1)(https://www.bioinformatics.babraham.ac.uk/projects/fastqc/) are available on the GitHub repository referred in the Data availability section. Then, the Nanopore files were assembled with Canu (v 1.6) [41] and polished with the Illumina reads obtained using Pilon (v. 1.22) [42]; three runs were necessary until no corrections were made anymore. Finally, the annotation was made with the classic RAST workflow available online [43].

## Genomic comparison and characterization of *E. coli* strains

In addition to BEN2908, described above, 22 other *E. coli* genomes used in this study were downloaded from NCBI's database under the following accession numbers and were also submitted to RAST for annotation: 55989 (GCF_000026245.1), 11368 (GCF_000091005.1), 042 (GCF_008042015.2), H10407 (GCF_000210475.1), 2009EL-2050 (GCF_000299255.1), 11128 (GCF_000010765.1), E2348/69 (GCF_000026545.1), E24377A (GCF_000017745.1), O157:H7 EDL933 (GCF_000732965.1), O157:H7 Sakai (GCF_000008865.1), IMT5155 (GCA_000813165.1), APEC O1 (GCA_000014845.1), IHE3034 (GCA_000025745.1), RS218 (GCA_000800845.2), NRG857c (GCA_000183345.1), LF82 (GCA_000284495.1), CFT073 (GCA_014262945.1), χ7122 (GCA_000307205.1), 78-Pyelo (GCA_014131615.1), UTI89 (GCA_000013265.1), SCU-397 (GCA_013358385.1), and K-12 MG1655 (GCA_000005845.2). Regarding extraintestinal strains, these comprised three APEC (APEC O1, IMT5155 and χ7122) [44–46], two NMEC (RS218 and IHE3034) [47,48] and three UPEC strains (UTI89, CFT073 and 78-Pyelo) [49–51]. Regarding intestinal strains, these comprised four enterohemorrhagic *E. coli* (EHEC; O157:H7 Sakai, O157:H7 EDL933, 11368 and 11128) [52–55], three enteroaggregative *E. coli* (EAEC; 042, 2009EL-2050 and 55989) [56–58], two enterotoxigenic *E. coli* (ETEC; H10407 and E24377A) [59,60], two AIEC (LF82 and NRG857c) [27,28], one enteropathogenic (EPEC; E2348/69) [61], and two commensal strains (SCU-397 and MG1655) [62,63]. Multilocus sequence type (MLST) and serotype information of these strains were described in the above cited publications and information was verified using data from the Enterobase database [64] and the following software: ClermonTyping (v. 24.02) [65], SerotypeFinder (v. 2.0) [66], MLST (v. 2.0) [5,67,68], and FimTyper (v. 1.0) [69]. Other programs, such as VFanalyzer (v. 6.0) [70], CRISPRCasTyper (v. 1.8.0) [71], MinCED [72], SecReT6 (v. 3.0) [73], PHASTEST (v. 3.0) [74], Roary (v. 3.13.0) [75] and KEGG (release 116) [76] were also used to complement this characterization. The KEGG Pathway (KP) and KEGG Orthology (KO) databases were used by mapping the KO assignment numbers of the uncharacterized ORF homologs identified in this work onto KP maps. This allowed us to predict metabolic pathways potentially related to the molecular functions of the novel ORFs. However, these predictions are based on homology-derived KO assignments and require experimental validation of protein activity, regulation and specificity. All programs, excepting Roary, were run using the .fasta files of the 23 strains on their respective web-based platforms using default parameters. Roary was executed in Linux Ubuntu 24.04.3 LTS terminal with default alignment configurations using the .gff files of each strain. The -r and -e parameters were used to generate R plots and align core genes using PRANK [77], respectively.

With the annotated genomes, Orthofinder (v. 2.5.4) [78] was used for the identification of orthogroups (i.e., groups of genes with an evolutionary relation). Afterwards, the files generated from this analysis were used for constructing an unrooted phylogenetic tree, described below.

 

## RAxML unrooted tree generation and average amino acid identity (AAI) to AIEC LF82

Phylogenetic tree construction was based on two files generated with Orthofinder: (i) a .txt file identifying all the single copy orthogroups (i.e., the identification of all groups of homologous genes that possess only one allele corresponding to each strain) and (ii) a .csv file containing all the Orthogroups found. Based on the information contained in these two files, Python scripts were made for the following operations: (i) generation of one .fasta file for each single copy orthogroup found; (ii) alignment of the homologous genes contained in each single copy orthogroup .fasta using MUSCLE (v.5) [79]; and (iii) concatenation of all the aligned genes in a single .multifasta file with one entry for each strain. After, trimAl (v. 1.5) was used to trim ambiguous regions in the alignment, resulting in 23 sequences of 987,155 amino acids, which were converted in a .phy file, using a custom Python script. Then, ModelFinder (from IQ-TREE v. 3.0.1) [80] was used to identify the best substitution matrix. After that, RAxML [81] was executed using a JTT substitution matrix incorporating a proportion of invariable sites and a gamma rate of heterogeneity. Node support was assessed with 1,000 nonparametric bootstrap replicates. To validate the tree topology, three additional phylogenies were generated under alternative substitution matrices and evaluated with bootstrap replicates. The trees and supporting files are available in the GitHub repository. Finally, the web tool iTOL (v.7)(interactive Tree of Life) [82] was used for tree visualization and image editing. To complement the result of the tree, the program EzAAI (v. 1.2.4) [83] was used to obtain AAI values and proteome coverage percentages of all strains to AIEC LF82.

## Ring image generation and CDS functional characterization

The ring comparison was generated using the software BRIG (BLAST Ring Image Generator; v. 0.95) [84]. The chromosomal ring comparison was made by setting BEN2908 as the reference genome against the genomes of the ExPEC/AIEC strains that clustered in the phylogenetic tree (grey arrows), excepting UTI89 and NRG857c, which presented the same alignment pattern of RS218 and LF82, respectively. These comprised two APEC (red tones; IMT5155 and APEC O1), one AIEC (orange; LF82), two NMEC (purple tones; RS218 and IHE3034), two UPEC (green tones; 78-Pyelo and CFT073), and one commensal *E. coli* (grey; K-12 MG1655). The plasmidial ring comparison was made by setting pBEN2908 as the reference and the sequences of four other ColV-like plasmids screened using Liu et al. criteria (2018) [85], i. e., if at least one gene from four or more of the following gene clusters were present: (i) *cvaABC* and *cvi* (the ColV operon), (ii) *iroB-CDEN* (the salmochelin operon), (iii) *iucABCD* and *iutA* (the aerobactin operon), (iv) *etsABC*, (v) *ompT* and *hlyF,* and (vi) *sitABCD*. The four plasmids comprised three APEC plasmids (red tones; p1ColV5155, pAPECO1Col-BM, and pAPEC-1) and one AIEC plasmid (orange; pO83-CORR). BLAST was run with BLAST+ (v. 2.12.0) using default BLASTn parameters (word_size = 11; reward = 2; penalty=−3; gapopen = 5; gapextend = 2; e-value = 10). Regions containing at least 4 kb that were absent (coverage values below 50%) in the commensal K-12 strain were tagged as genomic regions (GR) of interest. The 4 kb cutoff was chosen because it approximately corresponds to the size of small functional gene clusters (around 3−4 genes, given *E. coli* general gene density of 1 gene per kb) and because the same threshold has precedent in genomic descriptions of related APEC strains (APEC O1 and IMT5155), which aids comparative interpretation [44,45]. S1 Table shows a relation of the GC content, nucleotide coverage, and nucleotide identity of each GR from BEN2908 against the genomes of the other strains using BLASTn optimized for highly similar sequences (megablast). Each GR range drawn in the ring was determined by the end of the last CDS common to all strains until the beginning of the first CDS common to all strains. These limiting CDS are written in the "GR Limits" column in Table 3 and S2 Table. Further, the names of twenty-one previously identified regions, including four from BEN2908 genome (GimA, GimB, AGI-1, and AGI-3) [16,86,87] were also added to the ring comparison.

The predicted putative functions of some of the genes and operons within the GRs were verified using BLASTp and CD-Search (v 3.21) [88,89], and were classified as follows: Sugar metabolism (SM), Prophages (Phg), Metabolism of iron and other metals (Met), Secretion Systems (SS), Adhesion and Invasion (A/I), and Defense mechanisms (Def). The other

genes selected that did not correlate with these functions were grouped as General Metabolism (GM). The domains and genes selected on "CD-Search prediction" and "BLASTp identity" columns in Table 5 and S2 Table were, respectively, the hit result with an expect value closer to 0 and, preferentially, a reviewed entry from the UniProtKB database. Unreviewed entries were selected when an unreviewed homolog had a relevant mention in a previous paper (ORFs $3_6$-$5_6$ [90]; ORF $2_{34}$ [91]; ORF $5_{34}$ [92]; and ORFs $9_{34}$-$13_{34}$ [93] or when no reviewed entries were found (ORF $1_5$ and ORF $1_6$). Each novel ORF identified in this work or previously reported CDS or region were referred to as a "feature". For prophage-containing GRs, the boundaries were defined by the region between the attachment sites identified in PHASTEST, except for GRs 11 and 14, where no attachment sites were found. S2 Table contains a summary of the features and their characterization in all 36 GRs.

## Results and discussion

### Assembly, characterization and phylogenetic analysis of BEN2908

Sequencing analysis indicated that the genome of BEN2908 comprised the chromosome and a single plasmid. The chromosome contig has 5,061,728 bp and 50.6% GC content, and the plasmid has 133,673 bp and 49.8% GC content. The characterization of BEN2908 and the strains used in this study was conducted using the web-based programs from CGE, as described in the Methods section. Strain BEN2908 (phylogroup B2; serotype O2:H5) exhibits a characteristic STc95 APEC profile, similar to APEC strain IMT5155 (B2; O2:H5:K1) and APEC O1 (B2; O1:H7) (Table 1).

To compare the genome of BEN2908 to other *E. coli* strains, we generated an unrooted phylogenetic tree using the 3,101 common orthologues identified by Orthofinder (Fig 1). The strains used in this tree were selected to evaluate which typing scheme best clusters BEN2908 and to elucidate which pathogenic pathotypes show greater affinity to the AIEC strains LF82 and NRG857c. For those reasons, in addition to the three strains mentioned above, 20 model *E. coli* strains from different phylogroups, sequence types, serotypes, and pathotypes were selected. Also, considering the importance of type 1 fimbriae (T1F) to invasion, strains of different *fimH* types were also selected. Gene *fimH* is commonly used as T1F typing gene because it directly mediates attachment to host cells, and it has been shown that its allelic variation alters affinity for mannosylated receptors, affecting adhesion and invasion [94,95]. In the phylogenetic tree (Fig 1), BEN2908 clustered with other STc95 strains, especially close to APEC strain IMT5155 which has the same serotype (O2:H5:K1), but a different ST due to one allelic differentiation in the adenylate kinase housekeeping gene (adk55 rather than adk13). Another noticeable aspect of the tree is that the AIEC strains LF82 and NRG857c are more closely related to the eight ExPEC strains (grey arrows), including BEN2908, than to any other InPEC strains analysed in this work. This is further supported by the AAI values and proteome coverage analysis, which shows that LF82 proteins have more identity and are more covered if compared to these ExPEC strains than to any InPEC, excepting AIEC NRG857c. These results suggest that STc135 AIEC orthologs closely resemble those of the ExPEC strains that clustered with BEN2908 (also known for its invasive capacity), implying that some genomic features may underlie their pathogenic traits.

### Supplementary characterization using VFanalyzer, CRISPRCasTyper, Roary and PHASTEST and comparison to AIEC strains

The genetic proximity shown above is further supported by the results of additional characterization tools, such as CRISPRCasTyper and MinCED (for Cas typing and CRISPR spacers identification), VFanalyzer (for virulence related genes screening), and Roary (for core genome comparisons). CRISPR/Cas analysis revealed that, with the exception of CFT073, 78-Pyelo (which doesn't possess a CRISPR/Cas system) and χ7122, all ExPEC and both AIEC harbour the same Cas subtype (I-F), contrasting to the I-E subtype found in every InPEC and commensal strain (S3 Table). Additionally, excepting χ7122, at least half of the spacers from every ExPEC strain were identified in LF82 and at least one-third in NRG857c, contrasting with none found in common to InPEC or commensal strains (S4 Table). This suggests a genetic relationship between these AIEC and ExPEC, since strains that share a recent evolutionary history

**Table 1. Characterization of the strains used in this study[1].**

|  | Strains | Pathotype | Sequence Type | ST complex | Phylogroup | Serotype | fimH type |
|---|---|---|---|---|---|---|---|
| **ExPEC** | **BEN2908** | APEC | ST95 | STc95 | B2 | O2:H5:K1 | fimH2343[2] |
|  | **APEC O1** | APEC | ST95 | STc95 | B2 | O1:H7:K1 | fimH15 |
|  | **IMT5155** | APEC | ST140 | STc95 | B2 | O2:H5:K1 | fimH15 |
|  | **χ7122** | APEC | ST23 | STc23 | C | O78:H9:K80 | fimH35 |
|  | **RS218** | NMEC | ST95 | STc95 | B2 | O18:H7:K1 | fimH18 |
|  | **IHE3034** | NMEC | ST95 | STc95 | B2 | O18:H7:K1 | fimH18 |
|  | **UTI89** | UPEC | ST95 | STc95 | B2 | O18:H7 | fimH18 |
|  | **78-Pyelo** | UPEC | ST12 | STc12 | B2 | O21:H5 | fimH5 |
|  | **CFT073** | UPEC | ST73 | STc73 | B2 | O6:H1 | fimH10 |
| **InPEC** | **LF82** | AIEC | ST135 | STc135 | B2 | O83:H1 | fimH436 |
|  | **NRG857c** | AIEC | ST135 | STc135 | B2 | O83:H1 | fimH2 |
|  | **O157:H7 Sakai** | EHEC | ST11 | STc11 | E | O157:H7 | fimH36 |
|  | **O157:H7 EDL933** | EHEC | ST11 | STc11 | E | O157:H7 | fimH36 |
|  | **11368** | EHEC | ST21 | STc29 | B1 | O26:H11 | fimH440 |
|  | **11128** | EHEC | ST16 | STc29 | B1 | O111:H8 | fimH86 |
|  | **E2348/69** | EPEC | ST15 | – | B2 | O127:H6 | fimH57 |
|  | **H10407** | ETEC | ST48 | STc10 | A | O78:H11 | fimH41 |
|  | **E24377A** | ETEC | ST1132 | – | B1 | O-:H28 | fimH54 |
|  | **42** | EAEC | ST410 | STc23 | C | O-:H21 | fimH24 |
|  | **2009EL-2050** | EAEC | ST678 | – | B1 | O104:H4 | – |
|  | **55989** | EAEC | ST678 | – | B1 | O104:H4 | – |
| **Commensal** | **SCU-397** | Commensal | ST38 | STc38 | D | O86:H18 | fimH5 |
|  | **K-12 MG1655** | Commensal | ST10 | STc10 | A | O16:H48 | fimH27 |

[1]This analysis was conducted using the programs ClermonTyping, for defining the phylogroup, Enterobase, for defining the ST complex according to Wirth (2006) [5], and the characterization programs from the Center for Genomic Epidemiology (CGE), mentioned in the Methods section.

[2]The allele fimH2343 differs from fimH15 by a single nucleotide mutation (537 G>A), resulting in a non-synonymous substitution on the amino acid sequence (180 G>S).

often have identical spacers [96]. The presence of virulence and core genes also follow this trend, as a higher proportion of genes common to AIEC and ExPEC were found on both VFanalyzer and Roary. Roary results showed that AIEC strains have more genes in common to ExPEC (3,352) rather than InPEC (3,119) strains, yielding a difference of 233 genes (full output available on GitHub). VFanalyzer identified 85 virulence-related genes in both AIEC strains. Of these, six genes (toxins *vat* and *usp*, invasin *ibeA*, metal transport periplasmic binding protein *sitA*, and two T6SS components) were found exclusively in at least one ExPEC and absent in all other InPEC strain, in contrast to only three genes (long polar fimbriae genes *lpfBCE*) exclusively shared with InPEC strains (full output in S5 Table). Despite the small numeric difference, the genes shared exclusively with ExPEC are not restricted to a single locus as the *lpfBCE* genes, and their presence and functionality are usually related to ExPEC virulence traits, such as pyelonephritis, HBMEC invasion, and cytoskeletal modifications on different hosts cells [22,29,97–99]. In contrast, PHASTEST results diverged from the pattern observed, as AIEC most common phages were almost evenly distributed among InPEC and ExPEC strains (S6 Table).

## pBEN2908 is a ColV-like plasmid

Colicin V (ColV) plasmids have been shown to play a pivotal role in the pathogenesis of extraintestinal pathogenic *E. coli*. Its name was attributed to the production of colicin V, a bacteriocin originally described by Gratia in 1925 as "factor

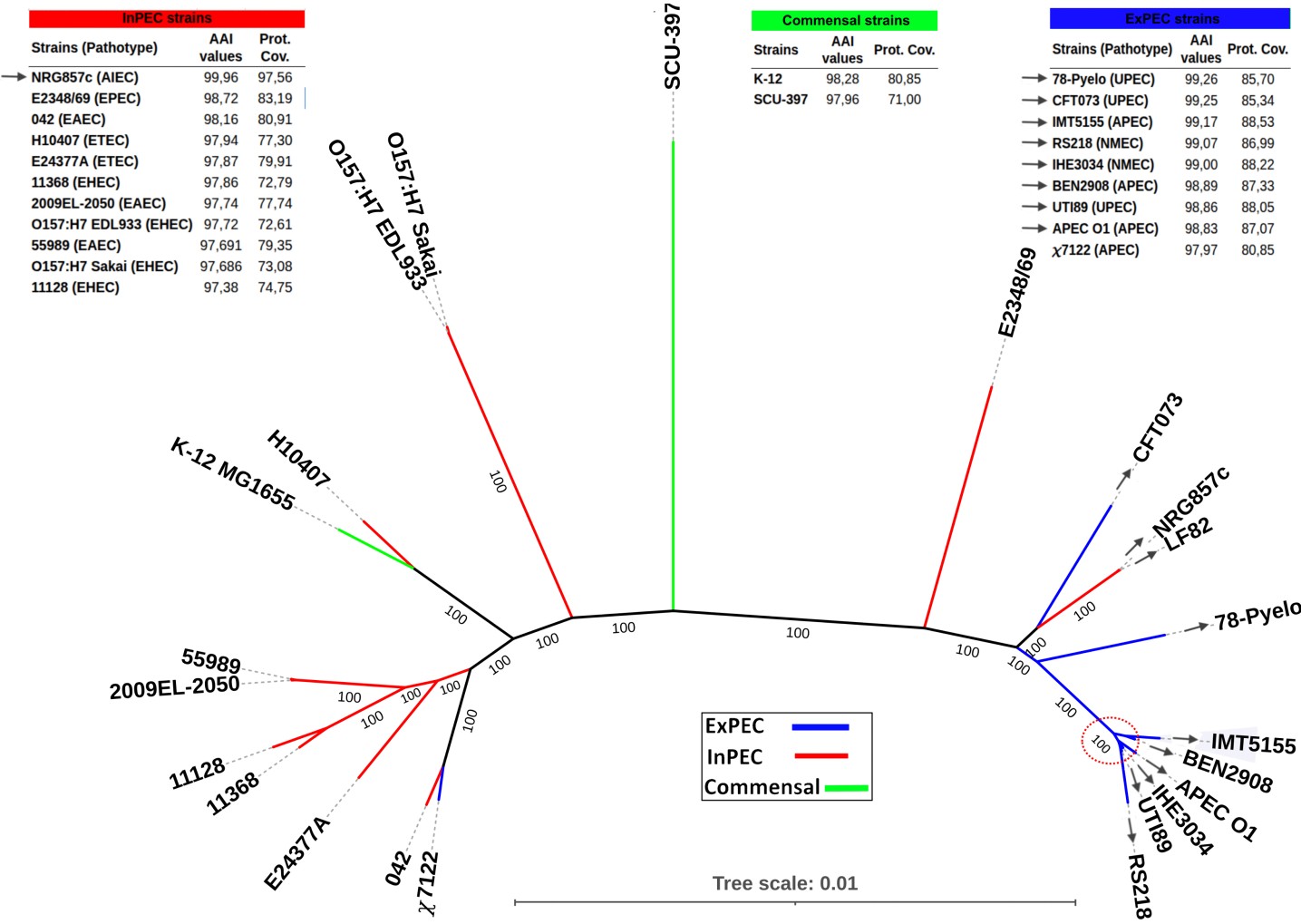

**Fig 1. Unrooted phylogenetic tree generated and AAI values of each strain to AIEC LF82.** Maximum likelihood phylogeny using the 3,101 common concatenated orthologues detected by OrthoFinder. The strains used are described in Table 1. Branches colored blue represent ExPEC, red represent InPEC, and green represent Commensal strains. The red dotted circle indicates that all nodes within have 100% bootstrap support. The grey arrows indicate the strains that clustered together and showed the highest AAI values and proteome coverage to AIEC LF82.

V" [100]. These plasmids usually range from 80 to 180 kbp and were identified in a variety of *E. coli* hosts, with particular significance in poultry infections [101–104]. Besides being widespread among hosts, the pathogenicity acquired from those plasmids is also noteworthy: acquisition of ColV-like plasmids by commensal/environmental *E. coli* from phylogroup B1 was associated with a more than threefold increase in infection rates among patients at a hospital in Paris [8,105]. This pathogenic potential relies on their gene content which encode various virulence factors, especially metal transport and uptake systems, such as the siderophores encoded by the aerobactin (*iuc/iut*) and salmochelin (*iro*) genes, but also the *sitABCD* and hypothetical *etsABC* metal transport systems [106]. The *iss* (increased serum survival) gene is also present on ColV plasmids and may contribute collectively with other systems to resist the bactericidal activity of serum complement [107]. Other genes such as outer membrane vesicle regulator *hlyF* and outer membrane protease *ompT*, type II toxin-antitoxin system *vapBC*, SOS inhibition system *psiAB*, and the conjugative transfer system *tra/trb* are also commonly found on these plasmids. To screen for ColV-like plasmids we applied the criteria defined by Liu et al. [85] to

the plasmids from 18 strains in our selection, as five strains (78-Pyelo, CFT073, 55989, MG1655, and SCU-397) did not harbour any plasmids. Of the 41 plasmids tested (description in S7 Table), only five fitted the criteria: four from the APEC strains (among which BEN2908) and one from the AIEC strain NRG857c (Table 2). To facilitate comparison, we generated a ring comparison containing only these five ColV-like plasmids and the ColV-associated genes described above with pBEN2908 as reference (Fig 2). The pLF82 is distinct from ColV plasmids, closely resembling the cryptic pHCM2 of *Salmonella enterica* serovar Typhi CT18 (isolated from a typhoid fever patient in Vietnam) [28]. This cryptic plasmid harbours phage-derived regions, a *parAB*-like partitioning module, a suite of DNA replication genes (including helicases, ligases, and exonucleases), and coding sequences similar to genes *rnhA* (ribonuclease H), *dhfR* (dihydrofolate reductase), *thyA* (thymidylate synthase), and *nrdAB* (ribonucleotide diphosphate reductase), involved in nucleotide synthesis [108]. Despite this difference, finding a ColV-like plasmid uniquely in an AIEC strain supports the idea that these plasmids may act as a zoonotic pathogenic trait derived from APEC [85].

**Table 2. Table of identities of ColV-like plasmids and the identity of their genes to the clusters from p1ColV5155[1].**

| | ST140 | ST95 | | ST23 | ST135 |
|---|---|---|---|---|---|
| Strain | IMT5155 | BEN2908 | APEC O1 | χ7122 | NRG857c |
| ColV-like plasmids (Acc. Number) | p1ColV5155 (NZ_CP005931.1) | pBEN2908 (LR740777.2) | pAPEC-ColBM (DQ381420.1) | pAPEC-1 (CP000836.1) | pO83_CORR (CP001856.1) |
| Plasmid size (bp) | 194,170 | 133,673 | 174,241 | 103,275 | 147,060 |
| PlasmidFinder best hit | IncFIB | IncFIB | IncFIB | IncFIB | IncQ1[2] |
| Cluster 1 (operon *iro*) | *iroN* (47259..49436) | 100 | 99.91 | 99.91 | 99.91 |
| | *iroE* (49481..50437) | 99.9 | 99.9 | 99.16 | 99.9 |
| | *iroD* (50522..51751) | 100 | 100 | 99.35 | 100 |
| | *iroC* (51855..55514) | 99.97 | 99.86 | 99.56 | 99.86 |
| | *iroB* (55654..56769) | 100 | 99.91 | 99.91 | 99.91 |
| Cluster 2 (operon *ets*) | *etsC* (68053..69423) | 100 | – | 99.83 | – |
| | *etsB* (70647..72143) | 100 | – | 99.83 | – |
| | *etsA* (72140..73327) | 100 | – | 99.83 | – |
| Cluster 3 (*hlyF* and *ompT*) | *ompT* (76295..77248) | 100 | 99.69 | 99.79 | 99.79 |
| | *hlyF* (77681..78790) | 100 | 99.73 | 99.73 | 99.73 |
| Cluster 4 (operon *sit*) | *sitA* (85429..86343) | 100 | 100 | 100 | 99.89 |
| | *sitB* (86343..87170) | 99.28 | 99.88 | 100 | 100[2] |
| | *sitC* (87167..88027) | 98.61 | 99.54 | 99.65 | 99.65 |
| | *sitD* (88024..88882) | 96.86 | 99.77 | 99.77 | 99.77 |
| Cluster 5 (operon *iut/iuc*) | *iucA* (92214..93938) | 100 | 100 | 100 | 100 |
| | *iucB* (93939..94886) | 99.89 | 99.79 | 99.79 | 99.79 |
| | *iucC* (94886..96628) | 100 | 100 | 100 | 100 |
| | *iucD* (96625..97958) | 100 | 99.78 | 100 | 100 |
| | *iutA* (97984..100185) | 100 | 99.91 | 99.91 | 99.86 |
| Cluster 6 (operon *cva*) | *cvaA* (166828..168069) | 100 | 99.76 | 99.76 | – |
| | *cvaB* (168062..170158) | 99.95 | 100 | 99.95 | 100[3] |
| | *cvaC* (170328..170639) | 100 | – | 100 | 99.36 |
| | *cvi* (170617..170853) | 100 | – | 100 | 99.16 |

[1]The plasmids shown are the ones from the 23 strains that fitted Liu et al. criteria [85] (see Methods section).

[2]Query not fully covered template length (529/ 796).

[3]11% alignment coverage.

 

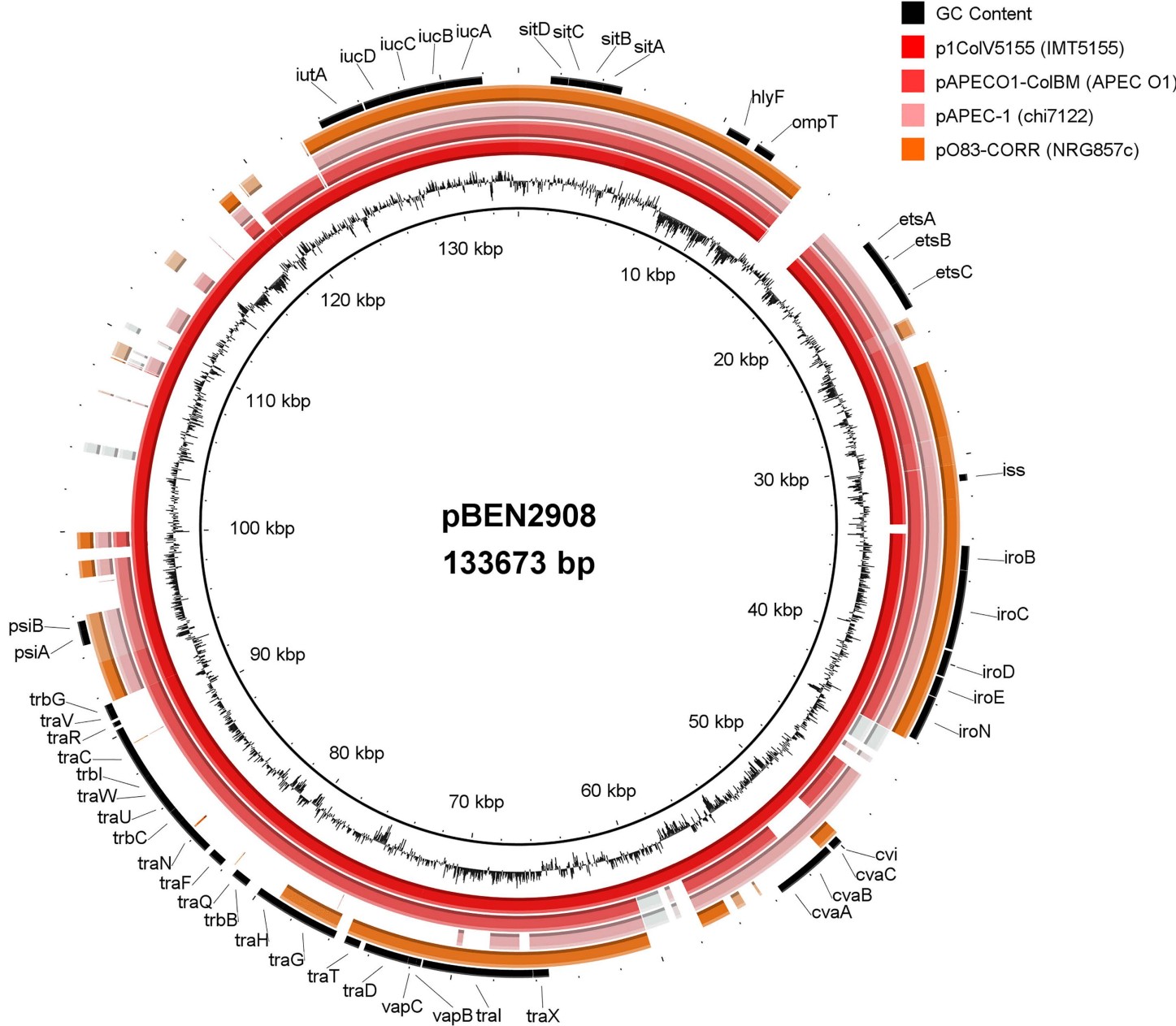

**Fig 2. Plasmid comparison using the software BRIG, setting pBEN2908 as the reference strain.** From the innermost to the outermost ring, the following plasmids are shown: p1ColV5155 (NZ_CP005931.1; IMT5155), pAPEC-O1-ColBM (DQ381420.1; APEC O1), pAPEC-1 (CP000836.1; χ7122), and pO83-CORR (CP001856.1; NRG857c). Genes usually present in ColV-like plasmids and the four gene clusters used in Liu et al. for ColV plasmid screening are shown in the figure [85].

## Overview and metabolic functions of the GRs in common between ExPEC and AIEC strains

While numerous genes have already been reported to contribute to virulence of extraintestinal *E. coli* [109,110], our research focused on identifying specific genomic regions possibly harbouring novel genes that could have a role in pathogenesis. Therefore, in an attempt to further analyse the genomic content of the strains that clustered with BEN2908

and AIEC LF82/NRG857c (Fig 1; grey arrows), we generated a ring comparison using the software BRIG. The BEN2908 genome was set as the reference and the following strains as subjects: two APEC (IMT5155 and APEC O1), two NMEC (RS218 and IHE3034), two UPEC (78-Pyelo and CFT073), one AIEC (LF82), and one commensal *E. coli* strain (MG1655).

The ring comparison identified 36 regions larger than 4 Kbp in the BEN2908 chromosome that were absent (below 50% of coverage) in the commensal strain MG1655 (Fig 3, S1 Table). To characterize the features within these regions, we used BLASTp and CD-Search. This analysis enabled us to classify the features into one or more of the following

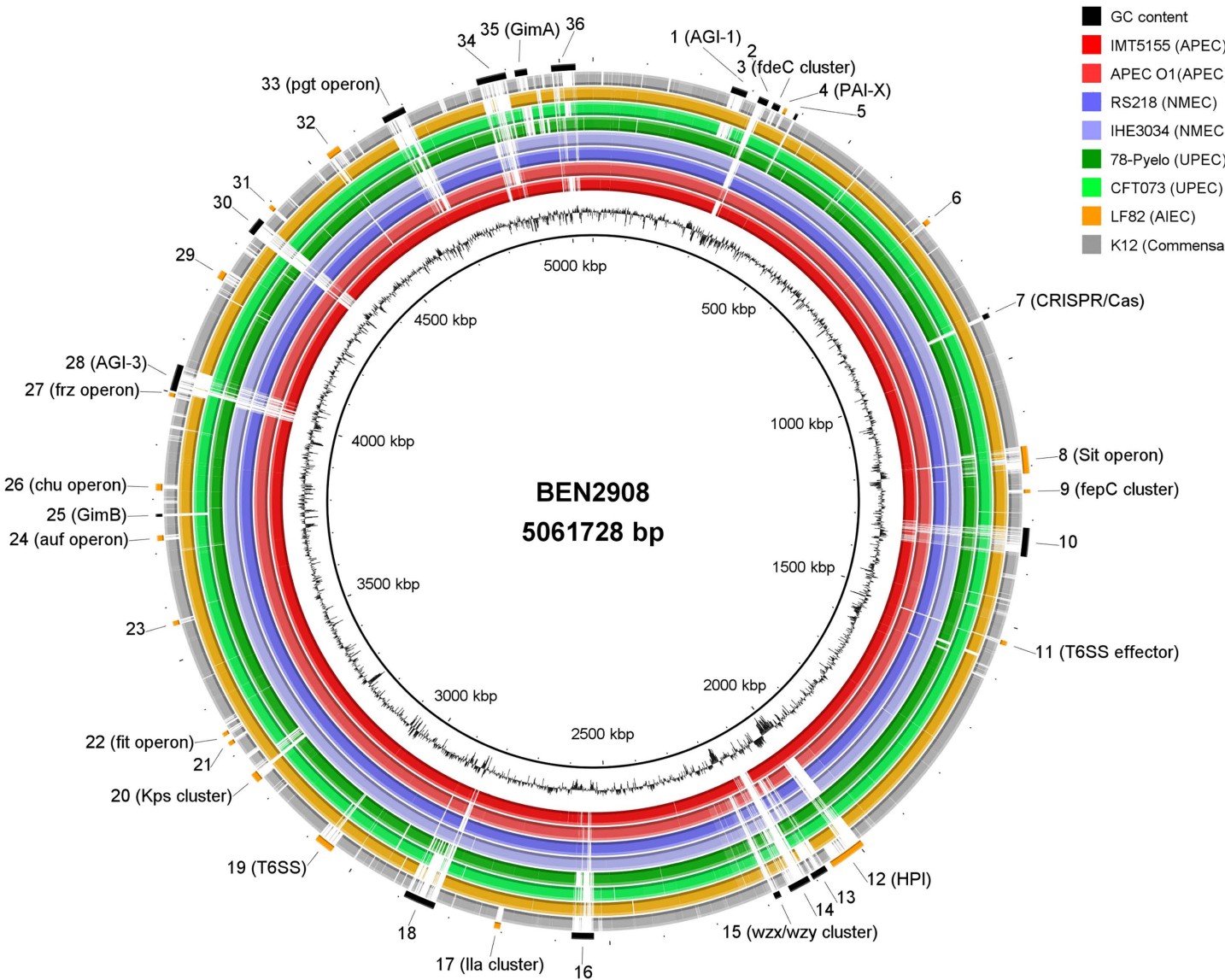

**Fig 3. Genome comparison using the software BRIG, setting BEN2908 as the reference strain.** From the innermost to the outermost ring, the following strains are shown: APEC IMT5155 and APEC O1 (red tones), NMEC RS218 and IHE3034 (purple tones), UPEC 78-Pyelo and CFT073 (green tones), AIEC LF82 (yellow) and commensal K-12 MG1655 (grey). Genomic regions (GRs) identified by the criteria described in the Methods section are indicated by numbers 1 to 36. The 20 regions marked in orange are common to all strains, except K-12 MG1655.

categories: "Sugar Metabolism" (SM), "Prophages," "Metabolism of Iron and other metals" (Met), "Secretion Systems" (SS), "Adhesion and Invasion" (A/I), and "Defense Mechanisms" (Def). Among these, 20 regions were also identified in all strains, excepting the commensal MG1655 with 15 exhibiting more than 70% coverage (Table 3, S1 Table). Features that did not fit into any of these categories were grouped as "General Metabolism" (GM). To better infer the functions of the uncharacterized features in this section, we used the KP and KO databases to investigate whether any pathway maps could be related to some of the hypothetical molecular functions of the ORFs identified. The following subsections describe the content of these 20 GRs - which are summarised in Table 4 (Reported features) and Table 5 (Uncharacterized features) – according to their respective categories, while also linking these findings to important ExPEC virulence traits.

**Sugar metabolism (SM). Reported.** The genomic regions 6, 19, 21, 23, 27 and 29 contain genes predicted to contribute to sugar metabolism (S2 Table), one of which (GR 27) has already been described. GR 27 contains a carbohydrate metabolic operon, named *frz* [18]. In this work, the authors showed that the presence of the *frz* operon promoted fitness, adhesion and internalization to different eukaryotic cell lines. Moreover, hybridization analysis of 151 ExPEC and 35 non-pathogenic avian *E. coli* strains showed that this operon is rarely present in non-pathogenic strains (5%) and its association increases with virulence, reaching 75% in the most virulent group [18].

**Uncharacterized.** The number of distinct GRs containing genes related to SM supports the importance of varied carbohydrate usage and uptake, enabling the bacteria to acquire various sugars that could be linked to other important

**Table 3. Genomic regions (GRs) of BEN2908 with more than 50% coverage to the ExPEC strains from this study and AIEC LF82.**

| GR | GR Limits | Start..Stop[1] | Completeness[2] | | | | | | |
|----|-----------|----------------|-----------------|--------|-----------|--------|-----------|--------|-------|
| | | | APEC O1 | IMT5155 | RS218[3] | IHE3034 | 78-Pyelo | CFT073 | LF82[4] |
| GR 3 | LSU rRNA L31p..*rclC* | 343380..357703 | T | T | T | T | T | T | T |
| GR 4 | *ykgH..betA* | 365342..371538 | T | T | T | T | T | T | T |
| GR 6 | *nagE..glnS* | 700554..710344 | T | T | T | T | T | T | T |
| GR 8 | *icd..ybcV* | 1162715..1213864 | T | T | T | T | T | T | P |
| GR 9 | *ymgE..treA* | 1243453..1250266 | T | T | T | T | T | T | T |
| GR 11 | *yncG..yddH* | 1528246..1536627 | P | T | T | P | T | T | P |
| GR 12 | *zinT..shiA* | 1992335..2062152 | P | T | P | P | T | T | P |
| GR 17 | *xseA..yfgJ* | 2704047..2715732 | T | T | T | T | T | T | T |
| GR 19 | tRNA-Met-CAT..*amiC* | 3049099..3085277 | T | T | T | T | T | T | T |
| GR 20 | tRNA-Phe-GAA..*yghD* | 3235081..3253327 | T | T | T | T | P | P | P |
| GR 21 | *ygiQ..ftsP* | 3318762..3326558 | T | T | T | T | T | T | T |
| GR 22 | *ygiN..parE* | 3338966..3346267 | T | T | T | T | T | T | T |
| GR 23 | *accC..yhdT* | 3563686..3572367 | T | T | T | T | T | T | T |
| GR 24 | *glpD..glgP* | 3723141..3733104 | T | T | T | T | T | T | T |
| GR 26 | *gor..yhiD* | 3816252..3828842 | T | T | T | T | T | T | T |
| GR 27 | *yicH..yicI* | 3991916..3999198 | T | T | T | T | T | T | T |
| GR 29 | *metE..ysgA* | 4228188..4244233 | T | T | T | T | T | T | T |
| GR 31 | *metL..metF* | 4388343..4396133 | T | T | T | T | T | T | T |
| GR 32 | *qorA..aphA* | 4533449..4560039 | T | T | T | T | T | T | P |

[1]Refers to the end of the gene immediately upstream and the beginning of the gene immediately downstream the GR.

[2]GRs with coverage values below 70% were considered partially (P) complete and were underscored and written in italic. GRs with more than 70% were considered totally (T) complete and were written in bold. All GRs had more than 95% sequence identity.

[3]RS218 and UTI89 have the same pattern. See S1 Table.

[4]LF82 and NRG857c have the same pattern. See S1 Table.

**Table 4. Summary of Genomic Regions (GRs) present in both ExPEC and AIEC strains from this study containing Reported features.**

| Category | GRs | Feature Occurrence (size) | Reported Features | Reference |
|---|---|---|---|---|
| **Sugar Metabolism** | GR 27 | 3991960..3999186 (7227 bp) | *frz* operon | [18] |
| **Metabolism of Iron and other metals** | GR 8 | 1208072..1211398 (3327 bp) | *sit* operon | [98] |
| | GR 9 | 1243503..1249884 (6382 bp) | *prrA-modD-yc73-fepC* cluster | [111] |
| | GR 12 | 1992335..2062152 (69818 bp) | HPI | [35] |
| | GR 22 | 3339011..3346203 (7193 bp) | *fit* operon | [112] |
| | GR 26 | 3819781..3828790 (9010 bp) | *chu* operon | [113] |
| **Adhesion and Invasion** | GR 3 | 344320..357543 (13224 bp) | *fdeC* cluster | [114] |
| | GR 4 | 365526..370467 (4942 bp) | PAI-X | [115] |
| | GR 17 | 2704116..2715523 (11408 bp) | *ila* cluster | [116] |
| | GR 24 | 3723195..3732610 (9416 bp) | *auf* operon | [117] |
| **Defense Mechanisms** | GR 20 | 3235318..3252268(16951 bp) | *kps* cluster | [118] |
| **Secretion Systems** | GR 11 | 1529090..1530652 (1563 bp) | T6SS Effector module 1 | [119] |
| | GR 11 | 1533136..1534173 (1038 bp) | T6SS Effector module 2 | |
| | GR 19 | 3049735..3079488 (29754 bp) | T6SS | [120] |
| **General Metabolism** | GR 29 | 4232707..4234704 (1998 bp) | *tkt1* | [121] |
| | GR 32 | 4540790..4542205 (1416 bp) | *dnaB* | [122] |
| | GR 32 | 4542258..4543337 (1080 bp) | *alr* | [123] |
| | GR 32 | 4544994..4546187 (1194 bp) | *tyrB* | [124] |

metabolic functions. Four genomic regions identified in the ring comparison (GRs 19, 21, 23 and 29) encode genes predicted to be related to the transport of carbohydrates (Table 5) such as Phosphotransferase systems (PTS: ORFs $4_{19}$ and $1_{29}$), ATP-binding cassette transporters (ABC-T: ORFs $3_{23}$, $4_{23}$, and $5_{23}$) and Tripartite ATP-independent periplasmic transporters (TRAP-T: ORFs $5_{21}$, $6_{21}$, and $7_{21}$) as well as probable transcriptional regulators (ORFs $5_{19}$, $1_{21}$, $7_{23}$, $4_{29,}$ and $5_{29}$). Additionally, with the exception of GR 21, hypothetical epimerases or isomerases were also present in those GRs (ORF $2_{19,}$ ORF $1_{23}$, and ORF $1_{29}$), likely playing a role in converting isomeric sugars into a form that can be metabolically processed by other enzymes following uptake.

Besides transport systems, additional enzymes potentially involved in carbohydrate processing were also identified. GR 6 harbours a pair of CDSs (ORFs $6_6$ and $7_6$) structurally analogous to the PdxA2 and DUF1537 protein families. This gene pair is involved in pyridoxal-5'-phosphate synthesis, an important cofactor for many enzymes such as alanine aminotransferases, which catalyze the reversible transfer of an amino group from alanine to 2-oxoglutarate to generate glutamate and pyruvate [90,125]. Some of the homologs of the identified ORFs in this region also cluster together in some bacteria. In *Sinorhizobium meliloti* 1021, for instance, enzymes structurally analogous to a sialidase (ORF $3_6$), a 4-hydroxy-tetrahydrodipicolinate synthase (*dapA;* ORF $4_6$), and an alcohol dehydrogenase (*adh;* ORF $5_6$), are encoded in a similar cluster. Additionally, *S. meliloti* carries an ABC transporter absent from the pathogenic *E. coli* strains. However, CD-Search of ORF $1_6$ returned a modest match to a citrate permease (Table 5, S2 Table), suggesting a possible transporter role that would require more robust confirmation [90].

Other features could also be noticed in these regions. Region 19, for instance, encodes a probable phosphoglycerate dehydrogenase (ORF $1_{19}$) and a beta-cystathionase (ORF $3_{19}$) with structural similarity to *malY*. MalY is capable of cleaving C-S linkages, producing central metabolic compounds such as pyruvate, and also represses the activity of the maltose regulon, being involved in catabolic regulation [126]. Additionally, in GR 23, a gene encoding a hypothetical tagatose aldolase (ORF $6_{23}$), showing high similarity to the catalytic GatY aldolase subunit from *E. coli* K-12 (63.95% identity) was identified. This enzyme plays an important role in glycolysis and gluconeogenesis, as it reversibly converts tagatose-1,6-bisphosphate to D-Glyceraldehyde-3P (G3P) and dihydroxyacetone phosphate (DHAP) [127]. Interestingly, by analysing

**Table 5. Genomic Regions (GRs) present in both ExPEC and AIEC strains from this study containing Uncharacterized features and their predicted identification.**

| Category | GRs | Feature Occurrence (size) | ORF number | CD-Search prediction (expect)[1] | BLASTp identity (Acc. code)[2] |
|---|---|---|---|---|---|
| **Sugar Metabolism** | GR 6 | 701923..700643 (1281 bp) | ORF $1_6$ | CitT (5.48e-24) | 88.50% to A0A285B4I9[3] |
| | | 702407..701937 (471 bp) | ORF $2_6$ | EbgC (3.73e-41) | 32.69% to NanQ (P45424) |
| | | 703573..702404 (1170 bp) | ORF $3_6$ | COG4692 (3.05e-138) | 46.34% to Q92X17[3] |
| | | 705816..704929 (888 bp) | ORF $4_6$ | DapA (2.68e-102) | 42.46% to Q92X22[3] |
| | | 706977..705820 (1158 bp) | ORF $5_6$ | Fe-ADH (4.41e-129) | 34.15% to Q92X15[3] |
| | | 707147..708382 (1236 bp) | ORF $6_6$ | OtnK (3.80e-126) | 40.38% to DtnK (Q8ZRS5) |
| | | 708375..709361 (987 bp) | ORF $7_6$ | PdxA (0.0) | 75.38% to PdxA2 (P58718) |
| | | 709393..710124 (732 bp) | ORF $8_6$ | GlpR (1.04e-89) | 32.50% to YgbI (P52598) |
| | GR 19 | 3080737..3079790 (948 bp) | ORF $1_{19}$ | PGDH_like_2 (6.92e-130) | 32% to SerA (P0A9T0) |
| | | 3081405..3080809 (597 bp) | ORF $2_{19}$ | GutQ (3.17e-88) | 52.43% to KdsD (P45395) |
| | | 3082583..3081408 (1176 bp) | ORF $3_{19}$ | MalY (1.04e-165) | 36.27% to MalY (P23256) |
| | | 3084112..3082583 (1530 bp) | ORF $4_{19}$ | PRK10110 (4.69e-155) | 41.02% to MalX (P19642) |
| | | 3085019..3084195 (825 bp) | ORF $5_{19}$ | PRK09772 (7.66e-28) | 30% to BglG (P11989) |
| | GR 21 | 3320086..3321558 (1473 bp) | ORF $2_{21}$ | MtlD (0.0) | 49.2% to UxuB (P39160) |
| | | 3321555..3322571 (1017 bp) | ORF $3_{21}$ | Zn_ADH7 (1.72e-163) | 55.3% to YjjN (P39400) |
| | GR 23 | 3564824..3563844 (981 bp) | ORF $1_{23}$ | Aldose_epim_Ec_c4013 (1.11e-145) | 32.86% to GalM (P0A9C3) |
| | | 3565783..3564821 (963 bp) | ORF $2_{23}$ | KdgK (4.07e-104) | 24.05% to KdgK (P37647) |
| | | 3566794..3565805 (990 bp) | ORF $3_{23}$ | AraH (4.09e-102) | 42.31% to RbsC (P0AGI1) |
| | | 3568294..3566795 (1500 bp) | ORF $4_{23}$ | MglA (0.0) | 44.06% to RbsA (P04983) |
| | | 3569245..3568355 (891 bp) | ORF $5_{23}$ | PBP1_ABC_sugar_binding-like (1.88e-108) | 32.07% to RbsB (P02925) |
| | | 3570135..3569281 (855 bp) | ORF $6_{23}$ | GatY (0.0) | 63.96% to GatY (P0C8J6) |
| | | 3570492..3571307 (816 bp) | ORF $7_{23}$ | GlpR (3.05e-50) | 32.60% to GlpR (P0ACL0) |
| | | 3571273..3572274 (1002 bp) | ORF $8_{23}$ | KdgK (3.75e-91) | 22.69% to KdgK (P37647) |
| | GR 29 | 4230549..4231106 (558 bp) | ORF $1_{29}$ | SIS_PHI (2.42e-80) | 39.44% to HxlB (P42404) |
| | | 4231148..4232656 (1509 bp) | ORF $2_{29}$ | PTS-II-BC-glcB (0.0) | 44.17% to PtsG (P69786) |
| | | 4235579..4234734 (846 bp) | ORF $4_{29}$ | RpiR (3.21e-74) | 18.33% to RpiR (P0ACS7) |
| | | 4236685..4235780 (906 bp) | ORF $5_{29}$ | PRK11074 (1.15e-95) | 22.03% to LysR (P03030) |
| **Defense Mechanisms** | GR 19 | 3081405..3080809 (597 bp) | ORF $2_{19}$ | GutQ (3.17e-88) | 52.43% to KdsD (P45395) |
| **General Metabolism** | GR 21 | 3319014..3319763 (750 bp) | ORF $1_{21}$ | PRK10225 (6.60e-60) | 24.68% to FadR (P0A8V6) |
| | | 3322582..3323592 (1011 bp) | ORF $4_{21}$ | AllD (1.29e-130) | 40.43% to AllD (P77555) |
| | | 3323665..3324648 (984 bp) | ORF $5_{21}$ | PBP2_TRAP_SBP_like_3 (2.06e-146) | 32.45% to DctP (P37735) |
| | | 3324690..3325172 (483 bp) | ORF $6_{21}$ | DctM (2.14e-32) | 30.11% to DctQ (O07837) |
| | | 3325183..3326487 (1305 bp) | ORF $7_{21}$ | DctQ (8.69e-144) | 36.30% to DctM (O07838) |
| | GR 29 | 4237986..4236763 (1224 bp) | ORF $6_{29}$ | SLC-NCS1sbd_CobB-like (9.35e-76) | 26.50% to CodB (P0AA82) |
| | | 4239374..4238418 (957 bp) | ORF $8_{29}$ | PRK12686 (0.0) | 47.16% to YqeA (Q46807) |
| | | 4240791..4239367 (1425 bp) | ORF $9_{29}$ | DUF1116 (5.27e-129) | 45.82% to YahG (P77221) |
| | | 4242347..4240788 (1560 bp) | ORF $10_{29}$ | PRK06091 (3.02e-165) | 40.48% to FdrA (Q47208) |
| | | 4243976..4243308 (669 bp) | ORF $12_{29}$ | PncA (1.92e-48) | 31.82% to RutB (P75897) |

*(Continued)*

**Table 5.** (Continued)

| Category | GRs | Feature Occurrence (size) | ORF number | CD-Search prediction (expect)[1] | BLASTp identity (Acc. code)[2] |
|---|---|---|---|---|---|
| | GR 31 | 4389421..4388561 (861 bp) | ORF $1_{31}$ | PRK15106 (2.14e-175) | 67.32% to Tsx (P0A261) |
| | | 4391117..4389498 (1620 bp) | ORF $2_{31}$ | UshA (3.32e-148) | 27.61% to YfkN (O34313) |
| | | 4392708..4391158 (1551 bp) | ORF $3_{31}$ | UshA (3.90e-152) | 27.52% to YfkN (O34313) |
| | | 4394198..4395751 (1554 bp) | ORF $4_{31}$ | UshA (1.09e-145) | 28.04% to YfkN (O34313) |
| | GR 32[4] | 4546369..4549113 (2745 bp) | ORF $11_{32}$ | SucA (0.0) | 48.76% to SucA/Odo1 (P0AFG3) |
| | | 4549146..4550300 (1155 bp) | ORF $12_{32}$ | PRK05704 (0.0) | 50.25% to SucB/Odo2 (P0AFG6) |
| | | 4550312..4551730 (1419 bp) | ORF $13_{32}$ | PRK06327 (0.0) | 38.65% to LpdA (P0A9P0) |
| | | 4551752..4552921 (1170 bp) | ORF $14_{32}$ | SucC (0.0) | 55.96% to SucC (P0A836) |
| | | 4552934..4553806 (873 bp) | ORF $15_{32}$ | SucD (0.0) | 67.59% to SucD (P0AGE9) |
| | | 4554018..4555517 (1500 bp) | ORF $16_{32}$ | Na_sulph_symp (2.55e-87) | 63.01% to Orf3 (Q07252) |
| | | 4555529..4556602 (1074 bp) | ORF $17_{32}$ | AllD (1.26e-112) | 44.88% to Ldh (Q07251) |
| | | 4557949..4556591 (1359 bp) | ORF $18_{32}$ | AtoC (0.0) | 52.26% to DctD (Q9HU19) |
| | | 4559762..4557942 (1821 bp) | ORF $19_{32}$ | PDC1_DGC_like (2.41e-10) COG4191 (1.23e-77) | 32.09% to DctB (Q9HU20) |

[1]specific hit or non-specific hit with expect value closer to 0.

[2]Preferentially reviewed entries were selected from the UniProtKB database, unreviewed entries were selected when the unreviewed homolog was previously mentioned in the literature (ORFs $3_6$, $4_6$, and $5_6$; [90]) or when no reviewed entry was found (ORF $1_6$).

[3]Unreviewed entry.

[4]Only the CDSs common to AIEC strains were shown, to see the complete annotation and characterization of GR 32, check S2 Table.

three KEGG pathway maps it was possible to trace a connection among ORFs $2_{21}$ and $3_{21}$ (GR 21), and $2_{23}$/$8_{23}$ (GR 23) that may also lead to the production of these compounds (Fig 4).

The predicted protein product of ORF $2_{21}$ presented homology to YjjN, an enzyme that converts L-galactonate to D-tagaturonate [128], which is a molecule present in both "Ascorbate and Aldarate metabolism" and "Pentose and Glucuronate interconversions" KEGG pathway. On this second map, the homolog of ORF $3_{21}$, named UxuB, catalyses the reversible interconversion of D-fructuronate to D-mannonate, and *vice-versa* [129]. By employing other enzymes present elsewhere on the genome (Fig 4, depicted in red: UxaA, UxaB, and UxuA), both reaction products (D-tagaturonate and D-mannonate) may be transformed to 2-dehydro-3-deoxy-D-gluconate (KDG). KDG is the substrate of the enzyme KdgK, whose sequences encoded by ORFs $2_{23}$ and $8_{23}$ showed significant structural homology containing all the specific ATP binding sites and active sites (S2 Table). This enzyme phosphorylates KDG, producing 2-dehydro-3-deoxy-D-gluconate-6P (KDPG), which is degraded to pyruvate and G3P in the final step of the Entner-Doudoroff pathway of glucose oxidation (Fig 4).

**Metabolism of Iron and other metals (Met). Reported.** For a bacterium to obtain iron within host extraintestinal tissues and fluids it requires specialized uptake systems called siderophores or other high-affinity systems to uptake iron-containing molecules such as haem or transferrin. Although the conserved siderophore enterobactin is the most widespread in *E. coli*, this siderophore is effective only in the intestinal lumen, as it is neutralized in extra-intestinal fluids by albumin and siderocalins [130]. Consequently, ExPEC lineages rely on a variety of patho-specific siderophores that evade host innate defense proteins such as siderocalin.

For that matter, different kinds of metal uptake systems contribute to the success of APEC strains in different types of infection, depending on the specific microenvironment inhabited by the pathogen [131]. All five regions (GRs 8, 9, 12, 22, and 26) contained previously reported ABC transporters and correlated metabolic enzymes: the *sit* operon (GR 8; related

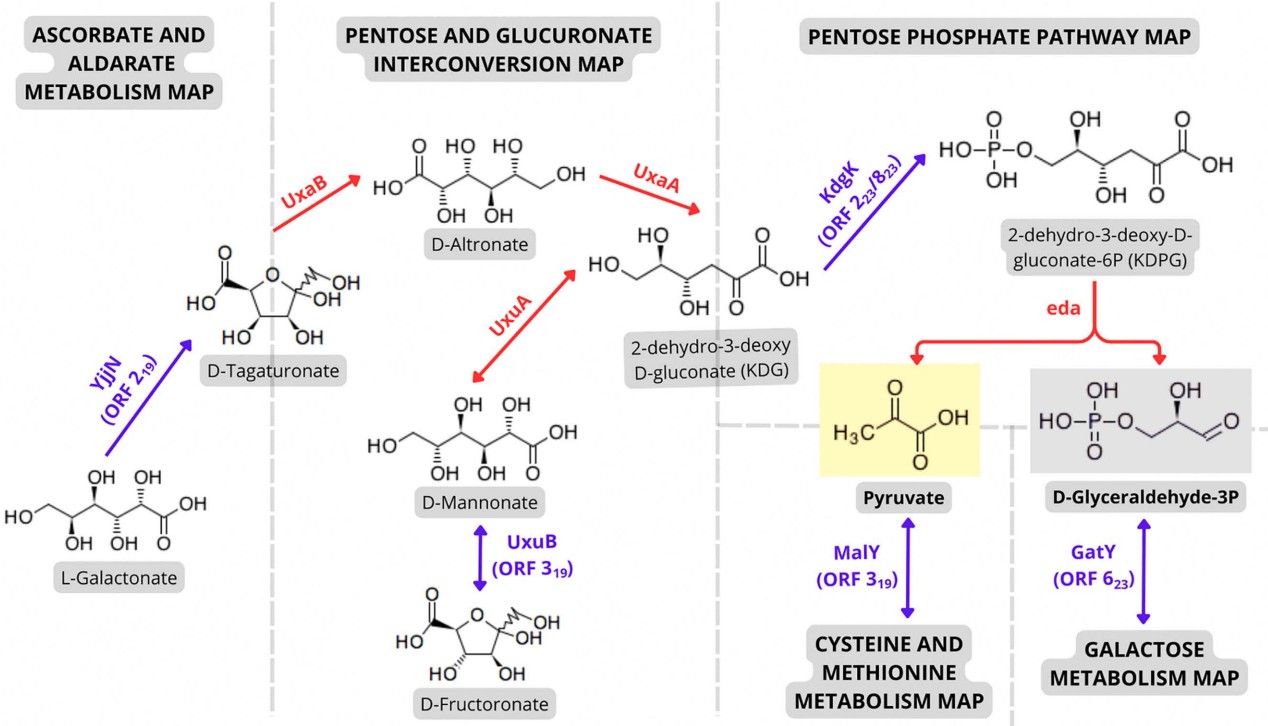

**Fig 4. Predicted sugar metabolism pathway based on KEGG maps and KEGG Orthology.** ORFs highlighted in purple represent putative novel coding sequences, while those in red correspond to genes located in other regions of the genome. ORFs from GRs 19, 21, 23 appear to be involved in the conversion of different sugars to KDPG, and, after, to Pyruvate and G3P, two important intermediates of glycolytic pathways.

to $Fe^{+2}$, but mainly $Mn^{+2}$ uptake [98]), the *prrA, modD, fepC, yc73* cluster (GR 9; possibly mediating iron uptake; [111]), the yersiniabactin cluster (GR 12; related to $Fe^{+3}$ and $Zn^{+2}$ uptake by the yersiniabactin siderophore system [132]), the *fit* operon (GR 22; related to $Fe^{+2}$, $Co^{+2}$, and $Cd^{+2}$ uptake [112]) and the *chu* operon (GR 26; related to the capture of haem through haemophores [113]). The roles of these systems for APEC infection were previously described in a number of reports [101,133,134]. For instance, it was shown that the most virulent APEC strains were able to grow in the presence of transferrin, in contrast to non-lethal strains, probably because these virulent strains contained additional iron uptake systems [135]. Also, it has been demonstrated that APEC O1 upregulated 13 genes related to metal uptake out of 20, when in contact with chicken serum and, among them were genes encoded by the *chu*, *ybt* (yersiniabactin), *sit* and *fep* operons [136].

**Adhesion and Invasion (A/I). Reported.** Four GRs previously reported in *E. coli* were linked to bacterial adhesion and invasion (GRs 3, 4, 17, and 24, Table 4). The capacity to adhere and invade is perhaps one of the main characteristics of strain BEN2908 [14,20–22]. An extensively studied factor involved in this capacity is the type 1 fimbria (T1F) encoded by the *fim* operon, present in most *E. coli* strains. This fimbria attaches to mannosylated host cell receptors, contributing directly to adhesion and invasion by various *E. coli* pathotypes [94]; T1F is considered a major virulence factor of ExPEC. Despite its conservation in most strains, some strains are unable to express T1F. T1F is regulated by a phase variation mechanism mediated by an invertible promoter switch, FimS, whose orientation can be flipped by a pair of recombinases called FimB and FimE [137]. While FimB inverts the *fim* switch in the on-to-off and off-to-on orientations with similar efficiencies, FimE inverts it rapidly in the on-to-off orientation [138]. Aside from these two recombinases, the gene *fimX*, present in GR 4, is an additional recombinase found in some strains. In the absence of the *fimE* and *fimB*, *fimX* also plays

a role in phase variation, turning the *E. coli* fimbriae rapidly to an "ON" state *in vivo* [139], further increasing adhesion and invasion properties as well.

In addition to T1F, the fimbrial system encoded by the *auf* operon, which has been linked to UPEC pathogenesis is present in GR 24 [140]. The *auf* operon is structurally similar to the *fim* operon, also containing: an adhesin (AufG), major (AufA) and minor subunits (AufE and AufD), two chaperones (AufB and AufF), and an usher (AufC) protein. The expression of this operon, evaluated by RT-PCR, was observed in strain CFT073 in at least three different times during infection of mice (4 h, 24 h, and 48 h post-infection) [117]. Aside from fimbrial systems, other genomic features are also linked to bacterial adherence [14,16,18,141]. Two other previously reported adhesin-related loci were detected: the gene encoding the *E. coli* adherence factor, FdeC (GR 3; [114]), and the Intimin-like Adhesin (Ila) encoding gene cluster (GR 17; [116]). Gene *fdeC* codes for a single large 1,416 residue protein with 95% identity to the surface protein EaeH, which was shown to promote adhesion [142]. It was shown by confocal micrographs that a strain lacking *fdeC* was incapable of adhering to bladder cells from the UM-UC-3 line, appearing to be involved in urinary tract infection (UTI) [114]. Moreover, the *fdeC* gene belongs to a locus containing eight other CDS, five structurally related to reductases and three putative regulatory genes. The ila cluster comprises three genes (*sinH*, *sinI*, *ratA*), and is probably derived from the genetic island CS54 identified in *Salmonella enterica,* which contains five genes, and is also linked to adherence and cell invasion. Many UPEC strains harbour the ila gene cluster, and it was shown that the deletion of these genes attenuated strains in a murine UTI model, due to reduced bladder cell invasion and decreased capacity to ascend the urinary tract, depending on the gene that was deleted [116].

**Defense mechanisms. Reported.** The modification of the bacterial cell surface is also a common defense strategy against host defenses and antimicrobial peptides. GR 20 harbours a genetic cluster known as the *kps* cluster, responsible for the synthesis and transport of components of the bacterial capsule [118]. The bacterial capsule is a polysaccharide-rich layer that surrounds the cell, playing a key role in evading host immune responses, mediating surface adhesion, and resisting desiccation. In *E. coli*, approximately 80 different capsule serotypes have been identified, highlighting the significance of capsule usage and adaptation [143].

**Uncharacterized.** Gram-negative bacteria have evolved a second external membrane that selectively allows compounds to enter through protein pores with size-exclusion properties or via diffusion across its hydrophobic lipid bilayer. Structures in the outer leaflet of this bilayer, such as lipopolysaccharides (LPS), play a critical defensive role. LPS is a modified lipid comprising three parts: Lipid A, the core oligosaccharide, and the highly variable O antigen, all of which can undergo modifications in response to environmental pressures [144]. For instance, Lipid A can be modified by the addition of 4-amino-4-deoxy-L-arabinose (L-Ara4N), a process likely mediated by ORF $2_{19}$ in GR 19, which shows high sequence homology with the KdsD arabinose isomerase responsible for this modification [145]. Such alterations can enhance bacterial resistance to positively charged particles, including cationic antimicrobial peptides (CAMP), thus contributing significantly to bacterial defense and survival.

**Secretion systems. Reported.** GRs encoding secretion systems belonged to the type 6 secretion system (T6SS), which are widely spread among Proteobacteria. The T6SS is a secretion apparatus that bacteria use to transport effector molecules that usually disrupt the targeted cell wall, being structurally related to a prophage injection apparatus. In *E. coli*, the T6SS has been described to be involved in interbacterial competition and in bacteria-host cell interactions, secreting different types of effectors, depending on the targeted structure [146]. In addition, many reports on *E. coli* have tested the role of T6SS in different *E. coli* pathotypes, being involved not only in the capacity to alter the structural actin filaments in human brain microvascular endothelial cells (HBMEC), but also to promote antibacterial effects through DNase activity [97].

To investigate the presence of T6SS genes in the strains investigated, we have employed the SecReT6 web resource [147]. Two GRs containing T6SS components could be identified: GRs 11 and 19. The latter contained more than fifteen CDS of this secretion system, being classified as T6SS-2, which is among the most prevalent sets found in APEC [97,148].

In addition to this cluster, the program SecReT6 identified two additional copies of the *tssI* (*vgrG*) gene in GR 11. One complete copy, with 1563 bp, exhibited 99.6% identity to the *tssI* from cluster 1 of APEC strain DE719 [149], while the smaller copy, with only 960 bp, showed 49.2% identity to the same gene. It's not unusual to find various copies of the *tssI*/*vgrG* homologs in the same strain, being reported in different gram-negative bacteria such as *P. aeruginosa* and *V. cholerae* [120,150]. For example, it has been shown that different VgrG homologs from *V. cholerae* strain V52 have distinct impacts on virulence, as only one of the three copies (VgrG-1) was able to cause modifications in cell actin structure. Moreover, these homologs have been found to interact with each other, forming various multimeric complexes that may affect the targeted cell differently. So, the presence of multiple copies of the *tssI*/*vgrG* gene in *E. coli* may have a potential for diverse functions, as demonstrated in other gram-negative bacteria.

**General Metabolism (GM).** The BRIG analysis identified four GRs associated with general or accessory metabolic genes present in the strains phylogenetically close to BEN2908 and AIEC LF82/NRG85. Of these, two (GRs 29 and 32) contained both reported and uncharacterized features, and two (GRs 21 and 31) consisted exclusively of uncharacterized features.

**Reported.** GR 29 is a 16 kb Genomic Island strongly associated with ExPEC from the phylogroup B2 and first identified in the APEC O1 genome [44]. In that island, a putative transketolase named Tkt1 (ORF $3_{29}$) showed 68% amino acid identity to TktA from *Vibrio cholerae* [121]. Notably, the Tkt1 protein could not complement the function of TktA involved in L-arabinose usage as a carbon source. Instead, it showed activity as an enzyme involved in peptide nitrogen extraction, since a mutant of this gene showed defects in the use of Pro-Ala or Phe-Ala as a nitrogen source. Interestingly, in GR 32, the genes *alr* and *tyrB* are both involved in the usage of some of the same residues tested by Li et al., since *alr* encodes an alanine racemase involved in the interconversion of both stereoisomers of alanine while *tyrB* encodes an aminotransferase that uses aromatic residues to transfer its amino group to 2-oxoglutarate and *vice-versa* [121]. In addition to these genes, GR 32 also harbours *dnaB* that encodes the extensively studied DNA helicase, which is the main replicative DNA helicase, participating in initiation and elongation during chromosome replication.

**Uncharacterized.** Other hypothetical enzymes encoded by genes in GR 29 (Table 5) also could be related to nitrogen obtention via carbamate hydrolysis. In the Rut pathway, nitrogen is obtained by cleaving the pyrimidine ring of a nitrogenous base, initially forming ureidoacrylate (product of RutA), followed by aminoacrylate and carbamate (product of RutB). Carbamate, in turn, is hydrolysed yielding nitrogen (as ammonia) and carbon (as $CO_2$) [151]. Notably, while ORF $12_{29}$ presented sequence homology to RutB, ORF $6_{29}$ showed similarity to a cytosine permease channel named CodB [152], indicating the potential presence of a pyrimidine transporter in this region. ORFs $8_{29}$ and $10_{29}$, in turn, presented homology to a carbamate kinase (YqeA) and to a oxamate carbamoyltransferase (FdrA), both capable of performing their reactions bidirectionally [153]. While FdrA transforms oxalurate to oxamate and carbamoyl phosphate, YqeA catalyses the transfer of phosphate from carbamoyl phosphate to ADP, forming ATP and leaving carbamate as a by-product, which, as mentioned earlier, is hydrolyzed.

Further expanding the network of nitrogen-related processes, the genes that encode hypothetical enzymes identified in GR 31 could be related to nucleotide processing or degradation. Despite displaying a modest (around 27%) amino acid identity to the *B. subtilis* 168 YfkN nucleotidase and no identity to the *E. coli* K-12 UshA nucleotidase, ORFs $2_{31}$, $3_{31}$, and $4_{31}$ each exhibited highly significant CD-Search matches to the conserved domains of UshA (Table 5; S2 Table). UshA is a 5'-nucleotidase that cleaves phosphate groups from nucleotides (with secondary NAD-pyrophosphatase activity), preserving all metal binding and active sites. Further, ORF $1_{31}$ exhibited high sequence identity to the nucleoside-specific channel Tsx, known to form pores for nucleoside uptake under substrate-limiting conditions [154,155]. Together, these findings could suggest the presence of a coordinated system for nucleoside degradation and transport, probably facilitating the processing of nucleosides.

Furthermore, other ORFs identified in GR 32 may have a role in the synthesis of tricarboxylic acid cycle intermediates, such as succinate. By plotting the KO number of the homologs of ORFs $11_{32}$, $12_{32}$, $13_{32}$, $14_{32}$ and $15_{32}$ in KEGG "Citrate Cycle (TCA Cycle)" map, it is noteworthy that they encode enzymes of a chain of reactions that lead to succinate

production from 2-oxoglutarate (Fig 5). These ORFs exhibited high protein identity and all the conserved functional sites to SucA, SucB, LpdA, SucC, and SucD, respectively (Table 5, S2 Table). Moreover, ORF $16_{32}$ showed significant similarity to the Orf3 permease from *Cupriavidus necator* H16, a dicarboxylate transporter responsible for the translocation of TCA cycle intermediates, including citrate, alpha-ketoglutarate, and succinate. This ORF also displayed the 206 conserved residues of the transmembrane helices of the Solute Carrier 13 permease domain, characteristic of these transporters [156]. Other ORFs also presented homology to dicarboxylate transporters, such as ORF $1_6$, which showed some structural similarity to CitT, a citrate transporter [157], and ORFs $5_{21}$, $6_{21}$, and $7_{21}$, mentioned above in the Sugar Metabolism subsection, that encode a hypothetical TRAP transporter, of which the most studied, *dctPQM* from *R. capsulatus*, is involved in C4-dicarboxylate uptake [158]. Notably, the two last ORFs from GR 32 presented similarity to the two-component regulatory system responsible for the expression of some dicarboxylate transporters. ORFs $18_{32}$ and $19_{32}$ presented significant homology to DctD and DctB from *P. aeruginosa* PAO1, respectively. When dicarboxylates are present, DctB autophosphorylates and transfers its phosphate group to DctD. DctD, in turn, binds to the promoter regions of DctA and DctPQM, which interacts with RpoN sigma factor, enabling its transcription and the posterior uptake of TCA cycle intermediates [159].

### Genomic rearrangements as influential forces for pathogenic *E. coli* adaptation

Some studies have shown that horizontal gene transfer (HGT) and gene duplication are important differentiation events in *E. coli* [160,161]. These events not only directly augment the arsenal of virulence genes, but also may give rise to new copies of existing genes that can evolve to acquire novel functions (neofunctionalization), complementary functions (subfunctionalization) or equal functions but with a different stimulatory network or a different expressing dosage [162].

**Fig 5. Predicted dicarboxylate utilization pathway based on KEGG maps and KEGG Orthology.** ORFs highlighted in purple represent putative novel coding sequences. Some ORFs from GR 32 appear to be involved in the conversion of 2-oxoglutarate to succinate, forming a second gene module analogous to the canonical *sucABCD* from *E. coli* K-12 MG1655.

Many GRs (6, 19, 21, 23, 29, and 32) contained ORFs that showed significant structural and sequence homology to known transporters (Table 5, S2 Table). Transporters are one of the most common categories seen in HGT and duplication events, because they are encoded by fewer genes than other systems, and possess fewer pleiotropic constraints, favouring their posterior fixation on the genome [163].

In our report, many of the GRs identified had gene modules that probably originated from HGT events and posterior gene rearrangements. In Fig 6A, for instance, the *pdxA*-like cluster from the pathogenic strains from this study is compared to one of the three *pdxA*-like clusters from *S. meliloti* 1021, described in [90] (Fig 2). Although the modular structure is different than any of the three from *S. meliloti* 1021, the sequence identity between five of the genes support the likelihood of a transfer event in the past. Curiously, *S. enterica* LT2 has a similar, but smaller cluster that was capable of complementing the function of the canonical *pdxA* gene *in vitro* (4PHT dehydrogenase) [90]. Notably, three of the four protein sequences from this cluster showed an even higher identity than *S. meliloti* 1021 (40.38% to ORF $6_6$, 75.38% to ORF $7_6$, and 71.17% to ORF $8_6$)(Table 5, S2 Table). This suggests that *Salmonella enterica* LT2 suffered enough selection pressure to discard some of the genes acquired, although more studies would be required to understand the correct timeline of events related to this cluster.

Fig 6B shows the high resemblance of ORFs from GR 29 to genes from *Photobacterium profundum* 3TCK, a resemblance significantly higher than to the homologous genes present on the *E. coli* K-12 genome (S2 Table). *Photobacterium* is a cosmopolitan genus of gram-negative marine bacterium that belongs to the family Vibrionaceae, and is present on the oceans in different depths, being considered piezophilic (optimal growth in high hydrostatic pressures) or non-piezophilic, depending on the species or even the strains. To live in different depths, each strain requires a series of adaptations to

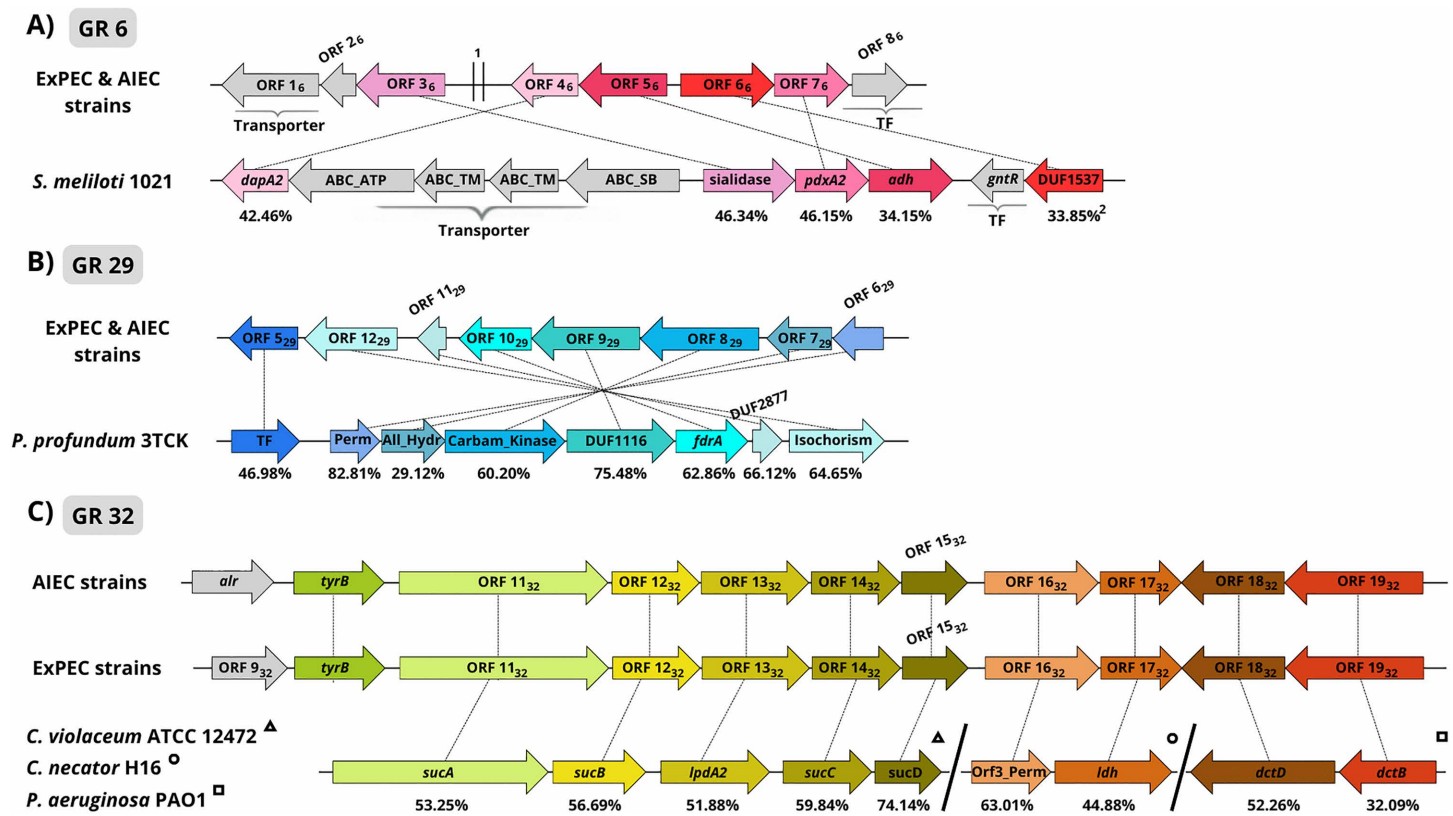

**Fig 6. Comparison of operons with gene organization similar to the observed in GRs 6, 29 and 32.**

different chemo-physical parameters, such as light, hydrostatic pressure, and organic carbon or nitrogen availability [164]. The strain *P. profundum* 3TCK is a non-piezophilic strain that was isolated from shallow waters in the San Diego Bay, it presents optimal growth under atmospheric pressure and at a broad range of temperatures (0 to above 20° C) [165]. The presence of this organism in shallow waters of a populous bay may have facilitated a transfer event to *Escherichia* at some point in the past [166]. Hypothetically, because this organism is from marine environments, the gene module shown on Fig 6B may be expressed to facilitate nitrogen acquisition through pyrimidine degradation (as mentioned in the GM subsection) when the *E. coli* strains are exposed to sea-like conditions (high salinity, colder temperature, etc). Curiously, it has been shown using a BIOLOG assay that another shallow water *Photobacterium*, named *P. marinum* J15, is capable of obtaining nitrogen through 41 (out of 95) different substrates, including the pyrimidines uridine and cytidine. Moreover, genomic analysis revealed that this strain also carries a carbamate kinase (similar to ORF $8_{29}$), which is likely involved in the nitrogen processing pathway [167].

Different from the cases depicted in Fig 6A (GR 6)in Fig 6B (GR 29),and in Fig 6C (GR 32), only 12 of the 19 ORFs identified in ExPEC were also present in AIEC strains. Among these, three are well-known genes: the helicase *dnaB* (ORF $6_{32}$), the alanine racemase *alr* (ORF $7_{32}$), and the aromatic aminotransferase *tyrB* (ORF $10_{32}$). Five of the missing ones occur between the quinone reductase, *qorA* (which is the GR upper limit) and *dnaB*, and the other two between *alr* and *tyrB* (for more information, see S2 Table). All the remaining ORFs (with the exception of ORFs $17_{32}$ and $19_{32}$) displayed more than 50% amino acid identity to homologous genes from different bacteria, suggesting that more than one genomic rearrangement has occurred throughout time. The first portion (Fig 6C, triangle), showed significant identity to the canonical *sucABCD* operon from *E. coli* K-12, although it had even greater similarity in sequence and operon arrangement to the *sucAB-lpdA2-sucCD* cluster from *Chromobacterium violaceum* ATCC 12472. *C. violaceum* is a gram-negative saprophyte found in water and soil of tropical regions. Nonetheless, it can also act as an opportunistic pathogen, with infections typically arising from contact with contaminated water or exposure of skin lesions to infected soil or water. This bacterium was already isolated in cases of bacteraemia, septicaemia, and UTI, with some infections leading to death [168]. Curiously, the second (Fig 6C, circle) and third portions (Fig 6C, square) of the GR 32 gene module are absent in *C. violaceum*. Instead, they show significant homology to a probable dicarboxylate permease and a lactate dehydrogenase from the gram-negative soil bacterium *Cupriavidus necator* H16, as well as to the two-component regulatory system *dctBD* of the TRAP transporter *dctPQM* from the gram-negative opportunistic pathogen *Pseudomonas aeruginosa* PAO1. This suggests that the genomic rearrangements within GR 32 have formed a "mosaic" gene module [155], and although some genes are absent in AIEC, the ORFs depicted in Fig 6C are conserved, preserving key modules regarding catabolism of dicarboxylates such as genes encoding proteins for transport (ORF $16_{32}$), processing (ORFs $11_{32}$, $12_{32}$, $13_{32}$, $14_{32}$, and $15_{32}$), and regulation (ORFs $18_{32}$ and $19_{32}$).

## Conclusion

Despite some differences such as the variation in phage-related regions (S6 Table), STc135 AIEC strains show strong genomic similarity to ExPEC strains, greater than with the InPEC included in the current genomic comparative analyses. This close phylogenetic relatedness (Fig 1), shared virulence gene profile (S5 Table), overlapping CRISPR spacers (S3 and S4 Tables) and greater core genome similarity support a recent common ancestry.

The ring comparison (Fig 3) revealed that of the 36 genomic regions of BEN2908 absent in *E. coli* K-12, 20 were also present on AIEC and ExPEC strains studied. Further investigation of these 20 regions reinforced aspects regarding ExPEC pathogenesis, such as the importance of iron and sugar uptake and metabolism (as supported by the number of GRs containing features related to those aspects, especially transporters), but also the importance of the T6SS (as effectors and cluster identified are commonly studied in APEC pathogenesis). In addition to that, the analyses of some of the GR features made with BLASTp, CD-Search, and KEGG databases indicated that ORFs related to nitrogen processing (identified in GRs 29, 31, and 32) and dicarboxylate uptake and metabolism (identified in GRs 6, 21, and 32) also appear

to be relevant traits in these *E. coli* lineages. Although deletion of some of the ORF homologs from those GRs resulted in decreased virulence in other bacteria, such as *sucA*/ORF $11_{32}$ in CFT073 [169], *sucABCD*/ORFs $11_{32}$, $12_{32}$, $14_{32}$, $15_{32}$ in *Salmonella* Typhimurium [170], *yqeA*/ORF $8_{29}$ in *E. coli* O102-ST405 [171], and *dctP*/ORF $5_{21}$ in *Vibrio alginolyticus* [172], none of these bacteria had a second copy of the affected genes, which is the case identified here. Moreover, the architecture of some of the genomic modules explored in this paper clearly resembles the architecture described for bacteria from other genera, with several genes presenting high amino acid identity, particularly those from GRs 29 and 32 (Fig 6) to those in other bacterial species. In spite of that, this is a predictive work and experimental validation for these *in silico* analysis is necessary to define the true functionality of the novel ORFs identified.

Finally, it is interesting to consider that the year of isolation of BEN2908 (1977) and the phylogenetically close strains studied spans decades, so the uncharacterized ORFs identified in this work (Table 5) remained conserved in those *E. coli* lineages, possibly indicating relevant roles for fitness, host adaptation, and virulence.

## Supporting information

**S1 Table. Table with GC content, coverage and identity.** [1] GC content calculated with the https://jamiemcgowan.ie/bioinf/gc_content.html web tool. [2] Bolded values have more than 70% coverage. Underscored values have between 50 and 70% coverage.
(XLSX)

**S2 Table. Table containing all GRs with reported and uncharacterized features.** [1] specific hit or non-specific hit with expect value closer to 0. Commented in the cells are the conserved residues identified by CD-Blast. [2] Sequencing made by Schouler and Trotereau (2016), available at: https://www.ncbi.nlm.nih.gov/nuccore/AY395687.1. [3] Unreviewed entries selected because of their mention in the following papers: ORFs $3_6$-$5_6$ (Thiaville et al., 2016); ORF $2_{34}$ (Majumdar et al. 2004); ORF $5_{34}$ (Gárcia-Sanchez et al., 2021); and ORFs $9_{34}$-$13_{34}$ (Lim et al., 2015). [4] Unreviewed entry. No reviewed entries found. [5] PHASTEST attL (2101049.2101064) and attR (2167393.2167408) are too upstream and downstream, respectively, encompassing genes present in all strains – even those lacking phage content. Therefore, the region boundaries were defined using the same criteria applied to the other GRs: end of the first gene common to all strains, beginning of the last gene common to all strains.
(XLSX)

**S3 Table. Table with information regarding spacers and type of CRISPR/Cas system using MinCED and CRISPR-Castyper.** [1] CFT073, 78-Pyelo and E2348/69 doesn't possess a CRISPR/Cas system.
(XLSX)

**S4 Table. Table containing the number of spacers shared in each strain.** [1] CFT073, 78-Pyelo and E2348/69 doesn't possess a CRISPR/Cas system. [2] Spacer sequence is present, but displayed one to four nucleotide insertions located at their 5′ or 3′ ends.
(XLSX)

**S5 Table. Table generated using VFAnalyzer (VFDB) with the 23 strains.** [1] Highlighted in any color is shown the 85 genes identified in AIEC strains by VFanalyzer, of which: Yellow represents the 55 genes present among at least one InPEC, one ExPEC, one Commensal, and LF82. Green represents the 18 genes present among at least one InPEC, one ExPEC, and LF82. Orange represents the 6 genes present between at least one ExPEC and LF82. Blue represents the 3 genes present between at least one ExPEC, one Commensal, and LF82. Red represents the 3 genes present between at least one InPEC and LF82.
(XLSX)

**S6 Table. PHASTEST summary statistics of the 23 strains from this study.**
(XLSX)

**S7 Table. Plasmids statistics.** [1] The strains IHE3034, CFT073, 55989, SCU-397, and K-12 do not harbour any plasmids. [2] Three plasmids from the strain χ7122 and the plasmid from LF82 were not available for download. [3] pLF82 information was extracted from Miquel et al. (2010) Table 2 and Supplemental S1 Table. [4] Small plasmid size, RAST annotation failed. Information obtained from Genbank and GC content calculator: https://jamiemcgowan.ie/bioinf/gc_content.html.
(XLSX)

## Acknowledgments

We thank Raul Simon Batista for assisting in editing the preliminary results.

## Author contributions

**Conceptualization:** Tobias Weber Martins, Fabiana Horn, Catherine Schouler.

**Data curation:** Angélina Trotereau, Maxime Branger.

**Formal analysis:** Tobias Weber Martins, Angélina Trotereau.

**Funding acquisition:** Fabiana Horn, Catherine Schouler.

**Investigation:** Tobias Weber Martins, Angélina Trotereau, Sébastien Houle, Charles M. Dozois, Daniel Brisotto Pavanelo.

**Methodology:** Tobias Weber Martins, Fabiana Horn, Catherine Schouler.

**Project administration:** Fabiana Horn, Catherine Schouler.

**Resources:** Fabiana Horn, Catherine Schouler.

**Supervision:** Charles M. Dozois, Fabiana Horn, Catherine Schouler.

**Writing – original draft:** Tobias Weber Martins.

**Writing – review & editing:** Tobias Weber Martins, Simone Iahnig-Jacques, Charles M. Dozois, Fabiana Horn, Catherine Schouler.

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
