## [Decision Letter · Decision Letter 0]

22 Oct 2025

PONE-D-25-49371Sequencing of the Invasive E. coli Strain BEN2908 Isolated from Poultry: A Comparative Investigation of Genomic Regions Shared with Other Invasive Model StrainsPLOS ONE

Dear Dr. Schouler,

Thank you for submitting your manuscript to PLOS ONE. After careful consideration, we feel that it has merit but does not fully meet PLOS ONE’s publication criteria as it currently stands. Therefore, we invite you to submit a revised version of the manuscript that addresses the points raised during the review process.

We look forward to receiving your revised manuscript.

Kind regards,

Feng Gao

Academic Editor

PLOS ONE

“This study has received funding from DGAL within the EcoAntibio2 call, COLIPHAVI project, France (C.S.), and from CNPq (Projeto Universal 423.902/2016‐4) and FAPERGS (PPSUS 21/2551-0000079-1), Brazil (F.H.). T.W.M. was the recipient of a CAPES Master studentship (DS 88887921543/2023-00). A.T. was supported by a training grant from the Fédération de recherche en infectiologie (FéRI). C.M.D. received funding from NSERC Discovery Grant 2019-06642.”

4. Thank you for stating the following in the Funding Section of your manuscript:

“This study has received funding from DGAL within the EcoAntibio2 call, COLIPHAVI project, France (C.S.), and from CNPq (Projeto Universal 423.902/2016‐4) and FAPERGS (PPSUS 21/2551-0000079-1), Brazil (F.H.). T.W.M. was the recipient of a CAPES Master studentship (DS 88887921543/2023-00). A.T. was supported by a training grant from the Fédération de recherche en infectiologie (FéRI). C.M.D. received funding from NSERC Discovery Grant 2019-06642.”

“This study has received funding from DGAL within the EcoAntibio2 call, COLIPHAVI project, France (C.S.), and from CNPq (Projeto Universal 423.902/2016‐4) and FAPERGS (PPSUS 21/2551-0000079-1), Brazil (F.H.). T.W.M. was the recipient of a CAPES Master studentship (DS 88887921543/2023-00). A.T. was supported by a training grant from the Fédération de recherche en infectiologie (FéRI). C.M.D. received funding from NSERC Discovery Grant 2019-06642.”

Reviewers' comments:

Reviewer's Responses to Questions

**Comments to the Author**

1. Is the manuscript technically sound, and do the data support the conclusions?

Reviewer #1: Yes

Reviewer #2: Yes

Reviewer #3: Yes

Reviewer #4: Yes

Reviewer #5: Partly

2. Has the statistical analysis been performed appropriately and rigorously? 

Reviewer #1: N/A

Reviewer #2: I Don't Know

Reviewer #3: N/A

Reviewer #4: Yes

Reviewer #5: N/A

3. Have the authors made all data underlying the findings in their manuscript fully available?

Reviewer #1: No

Reviewer #2: Yes

Reviewer #3: Yes

Reviewer #4: Yes

Reviewer #5: Yes

4. Is the manuscript presented in an intelligible fashion and written in standard English?

Reviewer #1: Yes

Reviewer #2: No

Reviewer #3: Yes

Reviewer #4: Yes

Reviewer #5: Yes

5. Review Comments to the Author

Reviewer #1: Peer Review: “Sequencing of the Invasive E. coli Strain BEN2908 Isolated from Poultry: A Comparative Investigation of Genomic Regions Shared with Other Invasive Model Strains” PONE-D-25-49371

Recommendation: Major Revision

Reviewer Comments

This is a very complete and carefully executed study. The authors present a full genome assembly and comparative analysis of the APEC strain BEN2908, with integration of plasmid features and comparison to other ExPEC and AIEC strains. The manuscript is generally well organized and within scope for PLOS ONE.

That said, to fully meet the journal’s standards of technical rigor, reproducibility, and cautious interpretation, I recommend major revision.

Global Comments

The manuscript would benefit from a clearer description of the origin of strain BEN2908. Please specify the sample type, host species, geographic origin, year of isolation, and whether this strain was previously described or deposited in a public collection. These details are critical to contextualize the comparative analysis with other ExPEC and AIEC isolates.

The genomic methodology is described in commendable detail, but essential elements are still missing to evaluate assembly quality and reproducibility. In particular,

The Introduction is well written and easy to follow, but the narrative would be strengthened by briefly introducing the main virulence and metabolic genes that are later discussed in the Results (e.g., iron uptake systems, T6SS loci, dicarboxylate transporters). Providing this context early on would prepare the reader for the subsequent comparative analysis.

In the Methods, the description of the Illumina sequencing is too limited. While the platform (MiSeq), read number, length, and GC content are reported, key information is missing for reproducibility and assessment of data quality.

Please, provide coverage statistics across the chromosome and plasmid, QC metrics (e.g., FastQC), and details on any trimming or filtering steps prior to assembly. Given that the data were generated in 2015, it would also be important to clarify whether quality was re-evaluated before use in this study. Finally, to meet PLOS ONE’s data availability requirements, the raw Illumina reads should be deposited in SRA with accession numbers included in the Data Availability Statement. Also, provide version/date for the web-based tools to ensure reproducibility (especially the tools that uses online databases).

The RAxML tree must include bootstrap support values. At present the conclusions about STc95 and STc135 are not supported.

Please specify the BLAST settings (identity, coverage thresholds, word size, or e-value) used to define the genomic regions (GRs), and justify the specific ≥4 kb cutoff, as this arbitrary threshold affects which loci are considered “GRs.”

In addition, while Tables S1–S2 summarize GC content, coverage, and identities, the underlying files (BRIG project, FASTA sequences of each GR, and CDS functional annotation tables) should be deposited in a public repository (for example Github, to allow replication). If it is available, scripts should be public on GitHub or public repositories, or more detailed in a section of supplementary methods. In the Table 2 and Figure they some plasmids are analyzed but they are not available for download. If pLF82 data were taken from an older publication, the comparative analysis is only partially reproducible; please provide the accession numbers or upload the sequence to a public repository.

For the functional assignments, please clarify the criteria for selecting “reviewed” vs “unreviewed” UniProtKB hits and provide the accession numbers of representative proteins. Without these details, other researchers cannot reproduce or verify the classification of features.

Please temper the VFAnalyzer/PHASTEST claims (from lines 221) by specifying identity/coverage thresholds, depositing the full outputs (Tables S4–S5 with raw hits), and avoiding absolute statements like “shares all” without qualification. I The same situation is in the plasmid section (from line 237) the description of pBEN2908 is super clear, but the statement that “all genes related to a ColV-like plasmid are present” is too absolute without showing the evidence in detail. A table listing each ColV-associated locus (e.g., iuc/iut, iro, etc) with accession, coverage, and identity values would be useful and also the Inc or MOB type groups in the same table.

In the Sugar Metabolism section (from line 328), please distinguish more clearly between functions previously validated in the literature (e.g., frz operon, GimA) and novel pathway predictions based on low-identity ORFs, so that readers can separate established evidence from putative annotations and similar in the identification of iron and metal uptake systems (from line 404), the phrasing “we identified systems related to metal uptake” should be revised to clarify that this refers to the detection of previously described operons through genome annotation, rather than novel experimental findings. In the Adhesion and Invasion section (lines 412), please clarify that operons and loci (fim, auf, fdeC, ila) were detected through annotation rather than experimentally demonstrated, and temper the functional claims for ORF 1₁₁ as putative predictions based on moderate homology.

The Discussion is very detailed and in parts can be excessive. Several subsections (e.g., Sugar Metabolism, General Metabolism) repeat pathway-level explanations that could be streamlined or moved to Supplementary Material. A more concise discussion, with selective emphasis on novel findings, would greatly improve readability. Consider moving some of the descriptive detail into the Introduction (to set context) or into Supplementary Notes, while keeping the main text focused on the comparative insights.

In the Conclusion, please temper the statement on long-term conservation of uncharacterized ORFs, which is only inferred from two isolates 21 years apart, and explicitly clarify that all proposed functions remain hypothetical pending experimental validation.

Minor revisons:

The statement that these findings “reinforce the importance in government monitoring of bacterial genomic evolution” (line 276) introduces a policy-oriented opinion that goes beyond the scope of the presented data. I recommend rephrasing this sentence to remain focused on the scientific evidence (i.e., the role of ColV-like plasmids in virulence).

Replace “softwares” with “software,” shorten long passive sentences.

Reframe causal or absolute statements (“shares all genes,” “ancestral proximity”) into cautious, data-supported wording (“shares X genes above threshold Y,” “phylogenetic proximity suggested but requires bootstrap support”).

Specific Comments

• Line 71: Please make explicit that you are now referring to human disease context.

• Line 93: Be careful with the term “cassette gene.” Are you referring to integron cassettes or simply modular groups of mobilizable genes? Clarify the terminology.

• Line 104: Explain or cite the “minor modification” of the ONT nanopore protocol. Without details, it is not reproducible.

• Line 120: Please, provide the RAST version

• Lines 113–115: Wording issue: “The obtained raw data (.fast5 files) was base called…” → should be “The raw data were basecalled with Guppy v4.0.11 using [config].” Please also specify qscore filter and parameters.

• Line 210–215: The statement that STc95 is closer to AIEC STc135 than to APEC SCI-07 cannot stand without bootstrap support. Please provide node support and, ideally, ANI/AAI values.

• Line 222–229: When describing shared virulence genes with LF82, avoid absolute terms like “all 76 genes.” Instead, specify thresholds for identity/coverage and include a full gene list in Supplementary Materials.

• Line 300–311 (Table 3): Provide coverage/identity values for each “genomic region” absent or present, or the cut off used for declare as presence/absence.

• Line 461–466: The CRISPR comparison is interesting, but please provide a full spacer table with coordinates and system subtype classification to avoid confusion.

• Line 650–656 (Conclusion): The wording “attests the utility and importance” is too strong. Suggest softening to “suggests possible functional utility, subject to experimental validation.”

Recommendation: Major revision.

This is a very complex and beautifully crafted manuscript that will be a valuable reference once revised. With the additions of assembly QC, reproducibility deposits, phylogenetic support, careful terminology and parameters, the paper will fully meet the PLOS ONE publication criteria. Once these issues are addressed, the manuscript will meet the journal’s standards for transparency, reproducibility, and cautious interpretation.

Reviewer #2: I see the main contribution of this paper serving as a potentially valuable resource, providing a comprehensive genomic comparison of BEN2908 and related E. coli strains, and identifying candidate loci for virulence that warrant experimental validation. The current manuscript is mainly descriptive. Below, I list some comments that need addressing before I believe it is suitable for publication.

Major Concerns

[1] Besides the strain of interest (BEN2908), 13 other E. coli strains are used to construct a phylogenetic tree. Instead of using the same 13 strains for the following analysis, the ring images are generated by comparing BEN2908 versus three other APEC strains, one AIEC strain (LF82), and one K-12 strain. Why only select a subset of the 13 strains for ring comparison?

[2] More details are needed to ensure the robustness of the phylogenetic analysis. When using MUSCLE, state whether ambiguous regions were trimmed and whether only the single-copy orthologs were included. For RAxML, report detailed bootstrap support values and/or compare the resulting topology under different substitution models.

[3] When constructing the phylogenetic tree, is it possible to incorporate some enteric pathotypes, such as E2348/69 (EPEC reference strain) or EDL933 (EHEC reference strain)?

[4] Around 23 genomic regions are identified in common between STc95 model strains and LF82. Since NRG867c is often compared with LF82 as representatives of AIEC, how many of these 23 genomic regions are also detected in NRG867c?

[5] When describing gene functions, I found it hard to distinguish functions predicted by KEGG from those demonstrated by experiments. The authors could emphasize functions that have been confirmed by both bioinformatics and bench.

Minor Comments

[6] Citation numbers 1 to 5 are missing in the main manuscript.

[7] Line 136, the software name “Orthofinder” should be “OrthoFinder”.

[8] Low figure resolution renders some key features unclear.

[9] The bibliography is extensive but not necessarily relevant. The authors should prioritize conclusions and related bibliography to emphasize the most novel findings.

Reviewer #3: Introduction

line 38/39- a general definition of avian colibacillosis is needed to help appreciate the significance of the work. I understand that it is not well defined, but perhaps just a general definition/classification of the disease... AC is a major cause of mortality, systemic bacterial infection, affects X% of poultry, costs to poultry farmers, something like that. The authors give slightly more information in the second paragraph, but it would help the reader to move some of the information to the first paragraph.

line 65- misplaced comma after "although". There are a few awkward sentences in the introduction and it should be re-read carefully and edited for clarity and ease of reading.

Line 87- if the paper is about both BEN2908 and LF82, why is BEN2908 the only strain mentioned in the title and why is LF82 not mentioned more in the abstract?

General comment: I think the introduction could be re-structured for better flow of information, especially for the benefit of genomic science/bioinformatics readers who are not e. coli experts. There are some instances where information is initially referenced, but the context is not added until later (example- initial mention of AIEC in line 63 by way of comparing BEN2908, but then an explanation of the importance/clinical context of AIEC isn't mentioned again until the next paragraph on line 71. Other examples: ST complex information in lines 38 and 42-47 could be put into better context- a brief mention of the difference between STc and ST would be helpful for readers who are not intimately familiar with E. coli, but are interested in the paper for the comparative genomic aspect- as could the relationship between AIEC and ExPEC strains (paragraph starting on line 71)).

Methods

Genomic comparison: pathotype acronyms should be defined. APEC and AIEC are previously defined, but NMEC and UPEC are not.

Results

Line 202: what is the rationale for only looking at fimH and not other components of type 1 fimbriae, such as FimA?

Line 208: the mention of x7122's sequence similarity to K12 feels distracting from the manuscript's goal of comparing BEN2908 to the rest and from the stated goal of the paragraph (in line 198).

Line 212: missing word "strains" after STc95 and before "(fumC38, mdh17, and recA26)".

Line 224-226: awkward sentence structure (and grammatical error) starting with "if considered (considering) exclusively".

Line 234: "results from the PHASTEST program went in the other direction" should be rephrased to clearly state the result shows a lack of commonality rather than using casual language and asking the reader to infer the result.

Line 238: please state the plasmid sizes in the text in addition to their location in the table.

Line 238: it might be interesting to also state what genes are found on the larger plasmids that are not found on pBEN2908.

Line 239: define and describe the characteristics of a ColV-like plasmid. Colicins are not mentioned in the text prior to this point, nor are they addressed/explained in this paragraph in a meaningful way.

Line 240: the authors state that "all genes related to a ColV-like plasmid are present on pBEN2908", but these genes are not explicitly highlighted in the text or in Figure 2. Please add an indication of which genes these are, either in the text or in the figure.

Line 241: "APEC plasmids are also known to encode various virulence factors" please confirm in the text that these are not only "known to be encoded" but ARE, in fact, encoded in the plasmid of interest.

Line 265: "pO83-CORR plasmid of strain NRG857c also carries all of the ColV‐associated genes" is this expected based on its ST, etc? Please note the ST or pathotype or such in the text so that the reader does not have to frequently flip back to Table 1 to look at the most important details and understand why this specific strain is being discussed for comparison.

Line 273: this level of significance would be great earlier on in this section of the results to help explain the significance of ColV plasmids. Include something like this up front to address my earlier comment on line 239.

Line 328: remove "the" at the beginning of the sentence. Start with "Genomic regions....". This also happens throughout the next few paragraphs- the authors switch back and forth between referring to GRs as "the GR #" or "GR #". Please choose one convention and stick with it, though I recommend removing "The" for ease of reading.

Line 336: The reference to "the authors showed" on this line is not clearly connected to citation #21.

Line 338-341: awkward sentence phrasing.

Line 352: please change "Aside from those" to more conventionally professional language, such as "Additionally".

Line 355: What is the significance of the catabolism of threonic acid to this paper that warrants its specific inclusion? Please elaborate. This paragraph in general is missing an explanation of the named genes' functions, unlike the surrounding paragraphs which do include that information.

Line 380: "pathway" should be plural.

Line 382: Potential rephrase, instead of using vice-versa: "...named UxuB, catalyses the reversible interconversion of..."

Line 426: The discussion of the interaction between Fim proteins does not include the function of FimH. Why, then, is this the Fim protein being analyzed (see also the comment on line 202)?

Conclusion

The limitations of the study should be acknowledged when drawing conclusions.

General comment: Casual and/or clunky language is used throughout the text. It should be carefully reviewed to ensure professionalism in the writing. Overall, the science is sound and the analysis is satisfactory.

Instances of "[author] and coll." should be edited to "[author] et al"

Reviewer #4: The manuscript “Sequencing of the Invasive E. coli Strain BEN2908 Isolated from Poultry: A

Comparative Investigation of Genomic Regions Shared with Other Invasive Model

Strains” by Martins et al., provide insight into sequence similarity between a invasive strain BEN2908 from poultry and compares it with previously reported E. coli genomes. The study identified sequence diversity as well as conserved regions in the genome and its biological significance. Following are the minor comments on the manuscript.

1. References for the genes that contribute to the virulence of E. coli (line 280) should be provided

2. In line 335, reference 21 appears in italics, and reference 153 is also italicized; please ensure consistent formatting.

3. The references numbered 154–159 are listed in the reference section but not cited in the main text.

4. At line 609, the temperature unit is incorrectly written as ‘Celsius’; it should be expressed in °C, consistent with the formatting used throughout the manuscript

5. Please provide a reference for the previously described GRs in E. coli mentioned in line 413.

6. Please include an appropriate reference to support the statement in line 531.

Reviewer #5: The authors have sequenced and compared BEN2908 genome with other model genomes of ExPEC strains and come up with some features that were unique to BEN2908, that is not shared with other genomes, that may contribute to pathogenesis.

Introduction: In the introduction, the authors describe the limited understanding of the pathophysiology of avian colibacillosis. In this context, they have chosen ST95 strains, particularly BEN2908, a well-studied strain. However, it is unclear why they also selected strain LF82—an adherent-invasive E. coli (AIEC) commonly associated with Crohn’s disease in humans—for comparison. Given that the focus of the study is the characterization of an extraintestinal pathogenic E. coli (ExPEC), what was the rationale for including a strain that is primarily an intestinal pathogen?

Methods: If the genome sequencing was outsourced to Genome and Transcriptome Facility at Bordeaux, France - technical details can be reduced to minimal.

The genome of BEN2908 strain was submitted in 2022 (The two accession numbers: LR740776.1; LR740777.2). This study has compared BEN2908 genome with 13 other genomes. The comparison was done with four APEC, two AIEC, two NMEC, three UPEC and two commensal E. coli strains. What was the rational to compare against just these categories?

Results & Discussion

The authors could have incorporated a larger set of genomes readily available from public databases, rather than limiting the analysis to just 14 genomes representing various ST types, pathotypes, and serotypes. This limited selection may reduce the resolution of the phylogenomic tree and the accuracy of virulome comparisons. The authors could have included a larger set of genomes and compared with a pangenome approach to get a wider and rather better picture on the virulome of strain BEN2908.

Fig 2 – Compares only plasmids of p1ColV5155 (IMT5155), pAPEC-O1-ColBM (APEC O1), and pAPEC-1( 7122) with that of BEN2908. What about plasmids from other genomes?

Page 21: lines 284-286 - The BEN2908 genome was set as the reference and five other strains as subjects: three model APEC strains (IMT5155, APEC O1, 7122), the commensal K-12 strain MG1655, and AIEC strain LF82. Why were only these genomes compared?

With the BRIG analysis, BEN2908was used as the reference genome and other genomes were compared to the reference genome. Comparing BEN2908 with other reference genomes would have given better insights of what BEN2908 was lacking.

6. PLOS authors have the option to publish the peer review history of their article (what does this mean? ). If published, this will include your full peer review and any attached files.

**Do you want your identity to be public for this peer review?** For information about this choice, including consent withdrawal, please see our Privacy Policy .

Reviewer #1: **Yes:** Florencia Martino, PhD

Reviewer #2: No

Reviewer #3: No

Reviewer #4: No

Reviewer #5: No

---

## [Author Response · Author response to Decision Letter 1]

18 Dec 2025

Response to Reviewers comments:

Reviewer #1: Peer Review: “Sequencing of the Invasive E. coli Strain BEN2908 Isolated from Poultry: A Comparative Investigation of Genomic Regions Shared with Other Invasive Model Strains” PONE-D-25-49371

Recommendation: Major Revision

Reviewer Comments

This is a very complete and carefully executed study. The authors present a full genome assembly and comparative analysis of the APEC strain BEN2908, with integration of plasmid features and comparison to other ExPEC and AIEC strains. The manuscript is generally well organized and within scope for PLOS ONE.

That said, to fully meet the journal’s standards of technical rigor, reproducibility, and cautious interpretation, I recommend major revision.

We thank the reviewer for these insightful and constructive comments, which have helped us to significantly improve our manuscript.

Global Comments

1. The manuscript would benefit from a clearer description of the origin of strain BEN2908. Please specify the sample type, host species, geographic origin, year of isolation, and whether this strain was previously described or deposited in a public collection. These details are critical to contextualize the comparative analysis with other ExPEC and AIEC isolates.

BEN2908 has been described in the introduction with the requested information (lines 70-73) and the following text was added in the Data availability section (lines 260-261) regarding deposition in a public collection: ”BEN2908 strain has been deposited at the International Center for Microbial Resources—Bacterial Pathogens (CIRM-BP) under name CIRMBP-1386.”

2. The genomic methodology is described in commendable detail, but essential elements are still missing to evaluate assembly quality and reproducibility. In particular, The Introduction is well written and easy to follow, but the narrative would be strengthened by briefly introducing the main virulence and metabolic genes that are later discussed in the Results (e.g., iron uptake systems, T6SS loci, dicarboxylate transporters). Providing this context early on would prepare the reader for the subsequent comparative analysis.

The Introduction was restructured to mention the importance of metal uptake in APEC infection, the recently shown relation of dicarboxylates in APEC virulence (lines 54-60), and the contribution of the T6SS to both APEC and AIEC pathogenicity (lines 102-106).

3. In the Methods, the description of the Illumina sequencing is too limited. While the platform (MiSeq), read number, length, and GC content are reported, key information is missing for reproducibility and assessment of data quality. Please, provide coverage statistics across the chromosome and plasmid, QC metrics (e.g., FastQC), and details on any trimming or filtering steps prior to assembly. Given that the data were generated in 2015, it would also be important to clarify whether quality was re-evaluated before use in this study. Finally, to meet PLOS ONE’s data availability requirements, the raw Illumina reads should be deposited in SRA with accession numbers included in the Data Availability Statement.

The requested information was added to the Methods “BEN2908 DNA extraction, sequencing, assembly and annotation” section (lines 143-151) and the sequencing files mentioned were submitted to GitHub, under the following repository: https://github.com/Martins-TW/BEN2908_Genome_Analysis.git

SRA files were submitted to BioProject PRJNA1359407 (lines 258-259).

4. Also, provide version/date for the web-based tools to ensure reproducibility (especially the tools that uses online databases).

All versions and releases were added to their respective Methods sections.

5. The RAxML tree must include bootstrap support values. At present the conclusions about STc95 and STc135 are not supported.

To improve phylogenetic support, we removed the draft genome from APEC SCI-07 and added 10 model InPEC strains. Our intent in this new selection is to show that AIEC STc135 and ExPEC strains phylogenetically close to BEN2908 have more similar orthologues than AIEC STc135 and InPEC strains. This is supported by the phylogenetic clustering of these strains, bootstrap values, Average Amino Acid Identity (AAI) comparisons, and consistent tree topology using different substitution matrices (available on github repository: link)

6. Please specify the BLAST settings (identity, coverage thresholds, word size, or e-value) used to define the genomic regions (GRs), and justify the specific ≥4 kb cutoff, as this arbitrary threshold affects which loci are considered “GRs.”

BLAST was run with BLAST+ using default blastn parameters (word_size=11; reward=2; penalty=−3; gapopen=5; gapextend=2; e-value=10). BRIG visualization thresholds were set to 90% (upper) and 70% (lower) identity; these thresholds control intensity colouring of the ring (lines 221-223). As suggested in the comment below, BRIG alignment files containing coverage and other data were uploaded at my GitHub account (link).

The ≥4 kb cutoff was chosen for the following reasons:

First, it corresponds roughly to 3-4 genes given the typical gene density in Escherichia coli (~1 gene per ~1 kb; some references on NCBI: K-12 (link), BEN2908 (link), LF82 (link). Thus, a 4 kb region is large enough to capture small functional clusters such as operons or adjacent co-functional genes, while still considering intergenic spaces and genetic elements that could not be identified by the characterization programs we used. Second, a 4 kb threshold has precedent in analogous genomic reports (e.g., genome announcements for APECO1 (link) and IMT5155 (link)), which facilitates comparison with closely related E. coli strains (lines 224-228).

To complement, we also note that some previously described genomic islands in E. coli (for example, GimB (GR 25) and PAI-X (GR 4)) have <5 kb in size (S2 Table). So, adopting a cutoff larger than 4 kb could exclude these biologically relevant islands.

7. In addition, while Tables S1–S2 summarize GC content, coverage, and identities, the underlying files (BRIG project, FASTA sequences of each GR, and CDS functional annotation tables) should be deposited in a public repository (for example Github, to allow replication). If it is available, scripts should be public on GitHub or public repositories, or more detailed in a section of supplementary methods.

The BRIG alignment files, FASTA sequences of each GR, their annotations generated by RAST, the python scripts used, as well as EzAAI, RAxML, Orthofinder, Roary, PHASTEST outputs and the sequencing files mentioned in methods section, are available on GitHub at the following repository (lines 257-259): https://github.com/Martins-TW/BEN2908_Genome_Analysis

8. In the Table 2 and Figure they some plasmids are analyzed but they are not available for download. If pLF82 data were taken from an older publication, the comparative analysis is only partially reproducible; please provide the accession numbers or upload the sequence to a public repository.

The section was revised so that the analysis no longer relies on plasmid comparison with pLF82, which is not available for download. The information we included about pLF82 in Supplementary Table S7 was extracted from the description provided in the supplementary material of Miquel et al. (2010).

9. For the functional assignments, please clarify the criteria for selecting “reviewed” vs “unreviewed” UniProtKB hits and provide the accession numbers of representative proteins. Without these details, other researchers cannot reproduce or verify the classification of features.

A better description of the reviewed vs unreviewed criteria was added to the “Ring Image generation and CDS functional characterization” Methods section (lines 243-247). The accession number of representative proteins from Uniprot database are available on Table 5 and S2 Table.

10. Please temper the VFAnalyzer/PHASTEST claims (from lines 221) by specifying identity/coverage thresholds, depositing the full outputs (Tables S4–S5 with raw hits), and avoiding absolute statements like “shares all” without qualification. I

PHASTEST full outputs were deposited in the GitHub repository and VFanalyzer full output is S5 Table with pathotype description and line coloring included. We also provided a description of how the programs were used in the “Genomic comparison and characterization of E. coli strains” Methods section (lines 179-186). The sentences were also modified to better attribute findings to the respective program outputs avoiding unqualified statements.

11. The same situation is in the plasmid section (from line 237) the description of pBEN2908 is super clear, but the statement that “all genes related to a ColV-like plasmid are present” is too absolute without showing the evidence in detail. A table listing each ColV-associated locus (e.g., iuc/iut, iro, etc) with accession, coverage, and identity values would be useful and also the Inc or MOB type groups in the same table.

As suggested, Table 2 now lists ColV-associated loci containing accession number, coverage, identity values, and PlasmidFinder identification. We also updated Figure 2 to include the same strains shown in Table 2.

12. In the Sugar Metabolism section (from line 328), please distinguish more clearly between functions previously validated in the literature (e.g., frz operon, GimA) and novel pathway predictions based on low-identity ORFs, so that readers can separate established evidence from putative annotations and similar in the identification of iron and metal uptake systems (from line 404), the phrasing “we identified systems related to metal uptake” should be revised to clarify that this refers to the detection of previously described operons through genome annotation, rather than novel experimental findings.

In the Adhesion and Invasion section (lines 412), please clarify that operons and loci (fim, auf, fdeC, ila) were detected through annotation rather than experimentally demonstrated, and temper the functional claims for ORF 1₁₁ as putative predictions based on moderate homology.

To clarify when the referred genes were already studied and when they were predicted, we divided each functional category (SM, Bf, A/I, …) into a Described and an Uncharacterized subsection. Also, we added a more careful statement when referring to low-identity ORFs, like ORF 16, mentioned in the Sugar Metabolism section (lines 466-470). GR 11 (so, ORF 111) was removed because we remade our alignments and found out that commensal K-12 have more than 50% of coverage and above 90% identity to GR 11. After this curation, 36 genomic regions remained absent from K-12 MG1655 and present in all strains from the ring (Fig. 3; S1 Table).

13. The Discussion is very detailed and in parts can be excessive. Several subsections (e.g., Sugar Metabolism, General Metabolism) repeat pathway-level explanations that could be streamlined or moved to Supplementary Material. A more concise discussion, with selective emphasis on novel findings, would greatly improve readability. Consider moving some of the descriptive detail into the Introduction (to set context) or into Supplementary Notes, while keeping the main text focused on the comparative insights.

As suggested, some of the text was removed from the discussion and some were modified to fit in the introduction.

14. In the Conclusion, please temper the statement on long-term conservation of uncharacterized ORFs, which is only inferred from two isolates 21 years apart, and explicitly clarify that all proposed functions remain hypothetical pending experimental validation.

As suggested, the statement was tempered and affirmation was broadened to clarify that these GRs are also present in several ExPEC strains, but require experimental validation to our predictive analysis (lines 769-772).

Minor revisions:

15. The statement that these findings “reinforce the importance in government monitoring of bacterial genomic evolution” (line 276) introduces a policy-oriented opinion that goes beyond the scope of the presented data. I recommend rephrasing this sentence to remain focused on the scientific evidence (i.e., the role of ColV-like plasmids in virulence).

Thanks for the comment, the suggested statement was removed.

16. Replace “softwares” with “software,” shorten long passive sentences.

done

17. Reframe causal or absolute statements (“shares all genes,” “ancestral proximity”) into cautious, data-supported wording (“shares X genes above threshold Y,” “phylogenetic proximity suggested but requires bootstrap support”).

Absolute statements were avoided, and wording was adjusted to reflect threshold-based interpretations.

Specific Comments

• Line 71: Please make explicit that you are now referring to human disease context.

added

• Line 93: Be careful with the term “cassette gene.” Are you referring to integron cassettes or simply modular groups of mobilizable genes? Clarify the terminology.

changed to genomic modules and gene modules

• Line 104: Explain or cite the “minor modification” of the ONT nanopore protocol. Without details, it is not reproducible.

Thank you for your comment. In fact, the minor changes are those indicated in the following sentences. The text has been amended as follows: Sheared DNA was End-Repaired using Oxford Nanopore recommendations for 1D Ligation sequencing (LSK-SQK 108), with minor modifications, as follows (line 129).

• Line 120: Please, provide the RAST version

done

• Lines 113–115: Wording issue: “The obtained raw data (.fast5 files) was base called…” → should be “The raw data were basecalled with Guppy v4.0.11 using [config].” Please also specify qscore filter and parameters.

The text was modified accordingly and the Guppy mode and qscore filter was specified (lines 139-142). The Nanoplot file generated comparing read quality before and after filtering was submitted to GitHub in the following repository: https://github.com/Martins-TW/BEN2908_Genome_Analysis

• Line 210–215: The statement that STc95 is closer to AIEC STc135 than to APEC SCI-07 cannot stand without bootstrap support. Please provide node support and, ideally, ANI/AAI values.

To improve phylogenetic support, we removed the draft genome from APEC SCI-07 and added 10 model InPEC strains. The new tree is supported by the phylogenetic clustering of these strains, bootstrap values, Average Amino Acid Identity (AAI) comparisons, and consistent tree topology using different substitution matrices (available on github repository; link)

• Line 222–229: When describing shared virulence genes with LF82, avoid absolute terms like “all 76 genes.” Instead, specify thresholds for identity/coverage and include a full gene list in Supplementary Materials.

VFanalyzer full output is S5 Table with pathotype description and line coloring included. We also provided a description of how the programs were used in the “Genomic comparison and characterization of E. coli strains” Methods section (179-186). As suggested the sentences were also modified to avoid absolute terms.

• Line 300–311 (Table 3): Provide coverage/identity values for each “genomic region” absent or present, or the cut off used for declare as presence/absence.

S1 Table contains coverage and identity values to the 22 strains and cutoff was explicit in S1 Table comment and Table 3 title (lines 420-424). Also, in “Overview and metabolic functions of the GRs in common between ExPEC and AIEC strains” Results and Discussion section (lines 398-400).

• Line 461–466: The CRISPR comparison is interesting, but please provide a full spacer table with coordinates and system subtype classification to avoid confusion.

S3 Table was built to contain the requested information.

• Line 650–656 (Conclusion): The wording “attests the utility and importance” is too strong. Suggest softening to “suggests possible functional utility, subject to experimental validation.”

As suggested, we rephrased the conclusion to adopt a more tempered tone as follows: “In spite of that, this is a predictive work and experimental validation for these in silico analysis is necessary to define the true functionality of the novel ORFs identified” (lines 766-768).

Recommendation: Major revision. This is a very complex a

---

## [Decision Letter · Decision Letter 1]

11 Jan 2026

PONE-D-25-49371R1Sequencing of the invasive E. coli strain BEN2908 isolated from poultry: a comparative investigation of genomic regions shared with intestinal and extraintestinal model E. coli strainsPLOS One

Dear Dr. Schouler,

Thank you for submitting your manuscript to PLOS ONE. After careful consideration, we feel that it has merit but does not fully meet PLOS ONE’s publication criteria as it currently stands. Therefore, we invite you to submit a revised version of the manuscript that addresses the points raised during the review process.

We look forward to receiving your revised manuscript.

Kind regards,

Feng Gao

Academic Editor

PLOS One

Journal Requirements:

Reviewers' comments:

Reviewer's Responses to Questions

**Comments to the Author**

1. If the authors have adequately addressed your comments raised in a previous round of review and you feel that this manuscript is now acceptable for publication, you may indicate that here to bypass the “Comments to the Author” section, enter your conflict of interest statement in the “Confidential to Editor” section, and submit your "Accept" recommendation.

Reviewer #1: All comments have been addressed

Reviewer #2: All comments have been addressed

Reviewer #3: All comments have been addressed

2. Is the manuscript technically sound, and do the data support the conclusions?

Reviewer #1: Yes

Reviewer #2: Yes

Reviewer #3: Yes

3. Has the statistical analysis been performed appropriately and rigorously? 

Reviewer #1: N/A

Reviewer #2: N/A

Reviewer #3: Yes

4. Have the authors made all data underlying the findings in their manuscript fully available?

Reviewer #1: Yes

Reviewer #2: Yes

Reviewer #3: Yes

5. Is the manuscript presented in an intelligible fashion and written in standard English?

Reviewer #1: Yes

Reviewer #2: Yes

Reviewer #3: Yes

6. Review Comments to the Author

Reviewer #1: Reviewer Report

I reviewed the revised submission together with the authors’ responses. Overall, the revision is clearly improved compared with the previous version. The manuscript reads more coherently, the methods are more reproducible (tool versions and key phylogenetic settings are now provided), and the comparative genomics results are presented in a clearer narrative. The authors have addressed most of the substantive concerns raised previously.

That said, there are still several points that should be corrected before acceptance.

Major points

1) Inconsistency in T6SS genomic region (GR) assignment.

In the “Secretion systems” section, the authors report T6SS in GR 11 and GR 19, but later state: “One set containing more than fifteen CDS … were found occurring in GR 20.” This does not match Table 4, where the larger T6SS cluster corresponds to GR 19 and modules are present in GR 11. Please verify and correct the GR numbering in the text (very likely GR 20 should be GR 19) so that the manuscript is internally consistent.

2) The Illumina read retention after filtering is extremely low and needs clarification.

The manuscript reports 32,274,430 MiSeq reads but only 1,713,025 reads retained after trimming and quality filtering. This is an unusually large reduction (approximately 95% discarded). This may reflect stringent filtering or an underlying quality issue, but the manuscript should explicitly state the filtering criteria (quality threshold, minimum length) and clarify whether counts refer to reads or read pairs. A short justification would help readers interpret this and reduce concern about data quality. Please clarify in methods the execution environment and provide version/parameters accordingly of “fastq_quality_filter”

3) Define explicitly what “absent in MG1655” means.

The ring comparison is built around calling genomic regions “absent in MG1655,” but the manuscript does not provide a clear operational definition of absence ( for example, coverage and identity thresholds). Since the authors already define inclusion thresholds in Table 3 (>50% coverage; “partial” <70%), the same explicit definition should be given for “absence. (pages 27 and 36)

Minor points

Page 24: “Read quality and length distributions were then assessed using.” Has an extra “.”

Page 36: (line 389) Double colon (“subjects: : two APEC”)

Page 29: Double period in Data availability (“LR740777.2. . Raw reads”)

Harmonize strain and plasmid naming conventions (APEC O1 vs APECO1). The manuscript alternates between “APEC O1” and “APECO1” (including in table headers). Please adopt a single convention for strain names and plasmid names and apply it consistently throughout.

Page 30: Table 1, rephrase the fimH2343 footnote for clarity. The note stating that fimH2343 results in the fimH5 translated amino acid sequence is difficult to interpret as written and mixes allele nomenclature with protein identity. Please rewrite this footnote more precisely (nucleotide change and resulting amino acid equivalence).

Page 44: Maintain consistent framing of homology-based inferences. In a few places, the wording still suggests experimental identification rather than annotation and or prediction. Minor edits (“annotated,” “predicted,” “detected operons involved in…”) would better align with the evidence (especially for metal uptake and pathway reconstruction).

Standardize the way the ring comparison strain set is counted. At one point the manuscript refers to “7 strains phylogenetically close,” while elsewhere it describes a comparison set including eight strains (plus a commensal reference). Please standardize the counting and wording to avoid confusion.

Add a brief limitation statement for the metabolic pathway reconstruction. The KEGG-based reconstruction is plausible, but it would be useful to add one sentence noting that this is inferred from homology-based KO assignments and requires experimental validation of regulation and transport specificity.

Recommendation

This is a strong revision and I think it can be accepted after the authors address the points above. The remaining issues are mainly about internal consistency (notably the T6SS GR numbering), interpretability/reproducibility (explicit thresholds; Illumina filtering and read counts), and copyediting (punctuation and terminology). Addressing these items will strengthen the manuscript and reduce the risk of additional questions in a subsequent round.

Reviewer #2: Thank you for addressing all my concerns and comments. Congratulations on the manuscript. I would appreciate it if the figures could be provided at a higher resolution.

Reviewer #3: (No Response)

7. PLOS authors have the option to publish the peer review history of their article (what does this mean? ). If published, this will include your full peer review and any attached files.

**Do you want your identity to be public for this peer review?** For information about this choice, including consent withdrawal, please see our Privacy Policy .

Reviewer #1: No

Reviewer #2: No

Reviewer #3: **Yes:** Katherine A. Innamorati, Ph.D.

---

## [Author Response · Author response to Decision Letter 2]

23 Jan 2026

Reviewer #1:

I reviewed the revised submission together with the authors’ responses. Overall, the revision is clearly improved compared with the previous version. The manuscript reads more coherently, the methods are more reproducible (tool versions and key phylogenetic settings are now provided), and the comparative genomics results are presented in a clearer narrative. The authors have addressed most of the substantive concerns raised previously.

That said, there are still several points that should be corrected before acceptance.

We thank you for your thorough review of our document, which has enabled us to correct the remaining errors

Major points

1) Inconsistency in T6SS genomic region (GR) assignment.

In the “Secretion systems” section, the authors report T6SS in GR 11 and GR 19, but later state: “One set containing more than fifteen CDS … were found occurring in GR 20.” This does not match Table 4, where the larger T6SS cluster corresponds to GR 19 and modules are present in GR 11. Please verify and correct the GR numbering in the text (very likely GR 20 should be GR 19) so that the manuscript is internally consistent.

Indeed, in this case GR 20 was GR 19, thanks for the comment. The text was modified accordingly. (line 613).

2) The Illumina read retention after filtering is extremely low and needs clarification.

The manuscript reports 32,274,430 MiSeq reads but only 1,713,025 reads retained after trimming and quality filtering. This is an unusually large reduction (approximately 95% discarded). This may reflect stringent filtering or an underlying quality issue, but the manuscript should explicitly state the filtering criteria (quality threshold, minimum length) and clarify whether counts refer to reads or read pairs. A short justification would help readers interpret this and reduce concern about data quality. Please clarify in methods the execution environment and provide version/parameters accordingly of “fastq_quality_filter”.

Thank you for your observation. Indeed the number was mistaken; it was actually 3,227,430 reads and not 32,227,430. To clarify each step of the processing, the text was modified to attend your considerations: “Illumina reads were trimmed for adapters and low-quality bases using Trimmomatic (v. 0.32)(38), with the following parameters: ILLUMINACLIP:TruSeq3-PE.fa:2:30:10; LEADING:30; TRAILING:30; HEADCROP:20; MINLEN:150. This resulted in 2,476,272 paired reads with minimum length of 150 bp, which were further filtered with a Q20 cutoff using fastq_quality_filter (v. 1.0.0), available on Galaxy platform (v. 25.0)(39). After filtering, a total of 1,713,025 reads were retained, yielding a coverage depth of 282x and breadth of 99.86%, as assessed by BWA (v. 0.7.19)(40).” (lines 148-153)

3) Define explicitly what “absent in MG1655” means.

The ring comparison is built around calling genomic regions “absent in MG1655,” but the manuscript does not provide a clear operational definition of absence ( for example, coverage and identity thresholds). Since the authors already define inclusion thresholds in Table 3 (>50% coverage; “partial” <70%), the same explicit definition should be given for “absence. (pages 27 and 36)

An explicit definition of absence was added to lines 239 and 411.

Minor points

Page 24: “Read quality and length distributions were then assessed using.” Has an extra “.”

done

Page 36: (line 389) Double colon (“subjects: : two APEC”)

done

Page 29: Double period in Data availability (“LR740777.2. . Raw reads”)

done

Harmonize strain and plasmid naming conventions (APEC O1 vs APECO1). The manuscript alternates between “APEC O1” and “APECO1” (including in table headers). Please adopt a single convention for strain names and plasmid names and apply it consistently throughout.

done, APEC O1 was defined as the convention.

Page 30: Table 1, rephrase the fimH2343 footnote for clarity. The note stating that fimH2343 results in the fimH5 translated amino acid sequence is difficult to interpret as written and mixes allele nomenclature with protein identity. Please rewrite this footnote more precisely (nucleotide change and resulting amino acid equivalence).

We agreed with your comment and the footnote was edited as follows:”¹ The allele fimH2343 differs from fimH15 by a single nucleotide mutation (537 G>A), resulting in a non-synonymous substitution on the amino acid sequence (180 G>S). (lines 293 and 294)

Page 44: Maintain consistent framing of homology-based inferences. In a few places, the wording still suggests experimental identification rather than annotation and or prediction. Minor edits (“annotated,” “predicted,” “detected operons involved in…”) would better align with the evidence (especially for metal uptake and pathway reconstruction).

To make a clear distinction between predicted and experimental identification we modified some paragraphs and made wording edits (lines 458-468, 476-479, 484-485, 531, 544, 566, 581, 601, 630, 632, and 786).

Standardize the way the ring comparison strain set is counted. At one point the manuscript refers to “7 strains phylogenetically close,” while elsewhere it describes a comparison set including eight strains (plus a commensal reference). Please standardize the counting and wording to avoid confusion.

To avoid confusion regarding the phylogenetically close strains we made wording edits (lines 311, 323, and 407).

Add a brief limitation statement for the metabolic pathway reconstruction. The KEGG-based reconstruction is plausible, but it would be useful to add one sentence noting that this is inferred from homology-based KO assignments and requires experimental validation of regulation and transport specificity.

We agree with your comment, and an explanation was added to the ”Genomic comparison and characterization of E. coli strains” Methods section: “The KEGG Pathway (KP) and KEGG Orthology (KO) databases were used by mapping the KO assignment numbers of the uncharacterized ORF homologs identified in this work onto KP maps. This allowed us to predict metabolic pathways potentially related to the molecular functions of the novel ORFs. However, these predictions are based on homology-derived KO assignments and require experimental validation of protein activity, regulation and specificity” (lines 187-192)

Recommendation

This is a strong revision and I think it can be accepted after the authors address the points above. The remaining issues are mainly about internal consistency (notably the T6SS GR numbering), interpretability/reproducibility (explicit thresholds; Illumina filtering and read counts), and copyediting (punctuation and terminology). Addressing these items will strengthen the manuscript and reduce the risk of additional questions in a subsequent round.

Reviewer #2:

Thank you for addressing all my concerns and comments. Congratulations on the manuscript. I would appreciate it if the figures could be provided at a higher resolution.

We thank the reviewer for the comment. Regarding the figures resolution, the PDF with the figures sent to you is generated by PLOS ONE during the submission process, and we have no control of the quality over this step. Nonetheless, as suggested, we improved the quality of the figures.

---

## [Decision Letter · Decision Letter 2]

29 Jan 2026

Sequencing of the invasive E. coli strain BEN2908 isolated from poultry: a comparative investigation of genomic regions shared with intestinal and extraintestinal model E. coli strains

PONE-D-25-49371R2

Dear Dr. Schouler,

We’re pleased to inform you that your manuscript has been judged scientifically suitable for publication and will be formally accepted for publication once it meets all outstanding technical requirements.

Kind regards,

Feng Gao

Academic Editor

PLOS One

Additional Editor Comments (optional):

Reviewers' comments:

Reviewer's Responses to Questions

**Comments to the Author**

1. If the authors have adequately addressed your comments raised in a previous round of review and you feel that this manuscript is now acceptable for publication, you may indicate that here to bypass the “Comments to the Author” section, enter your conflict of interest statement in the “Confidential to Editor” section, and submit your "Accept" recommendation.

Reviewer #1: All comments have been addressed

2. Is the manuscript technically sound, and do the data support the conclusions?

Reviewer #1: Yes

3. Has the statistical analysis been performed appropriately and rigorously? 

Reviewer #1: Yes

4. Have the authors made all data underlying the findings in their manuscript fully available?

Reviewer #1: Yes

5. Is the manuscript presented in an intelligible fashion and written in standard English?

Reviewer #1: Yes

6. Review Comments to the Author

Reviewer #1: Thank you for the careful and thorough revision. I re-read the revised manuscript and confirm that you have addressed the substantive points raised previously, including correcting the T6SS genomic region numbering, clarifying the Illumina read counts and filtering parameters (with versions and thresholds), and defining the operational criterion for “absence in MG1655”.

The manuscript is now internally consistent and substantially clearer and more reproducible, and the remaining issues are minor copyediting details that can be handled during production. On this basis, I recommend acceptance for publication.

7. PLOS authors have the option to publish the peer review history of their article (what does this mean? ). If published, this will include your full peer review and any attached files.

**Do you want your identity to be public for this peer review?** For information about this choice, including consent withdrawal, please see our Privacy Policy .

Reviewer #1: No

---

## [Editor Report · Acceptance letter]

PONE-D-25-49371R2

PLOS One

Dear Dr. Schouler,

I'm pleased to inform you that your manuscript has been deemed suitable for publication in PLOS One. Congratulations! Your manuscript is now being handed over to our production team.

Kind regards,

on behalf of

Dr. Feng Gao

Academic Editor

PLOS One